# Coupled ice/ocean interactions during future retreat of West Antarctic ice streams in the Amundsen Sea sector

David T. Bett[1], Alexander T. Bradley[1,2], C. Rosie Williams[1], Paul R. Holland[1], Robert J. Arthern[1] and Daniel N. Goldberg[3]

[1]British Antarctic Survey, Cambridge, UK
[2]Cambridge Zero, Cambridge, UK
[3]University of Edinburgh, Edinburgh, UK

*Correspondence to*: David T. Bett (d.bett@bas.ac.uk)

**Abstract.**

The Amundsen Sea sector has some of the fastest-thinning ice shelves in Antarctica, caused by high, ocean-driven basal melt rates, which can lead to increased ice stream flow, causing increased sea level rise (SLR) contributions. In this study, we present the results of a new synchronously coupled ice-sheet/ocean model of the Amundsen Sea sector. We use the WAVI ice sheet model to solve for ice velocities and the MITgcm to solve for ice thickness and three-dimensional ocean properties, allowing for full mass conservation in the coupled ice/ocean system. The coupled model is initialised in the present day and run forward under idealised warm and cold ocean conditions with a fixed ice front. We find that Thwaites Glacier dominates the future SLR from the Amundsen Sea sector, with a SLR that evolves approximately quadratically over time. The future evolution of Thwaites Glacier depends on the life-span of small pinning points that form during the retreat. The rate of melting around these pinning points provides the link between future ocean conditions and the SLR from this sector and will be difficult to capture without a coupled ice/ocean model. Grounding-line retreat leads to a progressively larger Thwaites ice-shelf cavity, leading to a positive trend in total melting, resulting from the increased ice basal surface area. Despite these important sensitivities, Thwaites Glacier retreats even in a scenario with zero ocean-driven melting. This demonstrates that a tipping point may have been passed in these simulations and some SLR from this sector is now committed.

## 1 Introduction

The West Antarctic Ice Sheet (WAIS) is a marine ice sheet of particular importance for future sea level rise (SLR), contributing ~6.5 mm to global SLR between 1992 to 2021 (Otosaka et al., 2023). Within the WAIS, the Amundsen Sea sector has seen the largest SLR contribution over the satellite era (Shepherd et al., 2019). The region experienced a 77% ice mass flux increase from 1973 to 2013, with Thwaites Ice Shelf specifically observing a 33% speedup during this period (Mouginot et al., 2014).

The Amundsen Sea sector's ice sheet is grounded on bathymetry below sea level, and the presence of retrograde slopes in the bathymetry may make the region susceptible to rapid and sustained retreat in the future (Favier et al., 2014). Any reduction in ice shelf buttressing via pinning points and side drag can lead to acceleration in ice shelf speed and retreat of the grounding line (Thomas, 1979). Ocean melting can reduce buttressing by thinning ice shelves, reducing side drag and aiding the ungrounding of pinning points. Of particular concern is Thwaites Glacier, which has a largely unconfined ice shelf, but whose

current pinning point on its eastern ice shelf appears to be weakening and has been predicted to unground within decades (Wild et al., 2022).

In the Amundsen Sea, warm modified Circumpolar Deep Water (CDW) resides below a colder and fresher layer of Winter Water (Jacobs et al., 1996), and flows through bathymetric troughs to the base of ice shelves (Walker et al., 2007), causing

high melt rates. The region experiences large decadal variability in its ocean conditions, most notably in the thickness of the modified CDW layer (Jenkins et al., 2018; Dutrieux et al., 2014). In addition, it has been suggested that there is an average anthropogenic warming trend superimposed on this internal variability (Holland et al., 2022; Naughten et al., 2022). Future anthropogenic warming of the Amundsen Sea is a key mechanism by which human activities may influence SLR from the Antarctic Ice Sheet (Holland et al., 2022; Jourdain et al., 2022; Holland et al., 2019).


Previous studies have used uncoupled ice-only simulations to simulate the future evolution and retreat of the Amundsen Sea sector and WAIS (e.g., Yu et al., 2018; Alevropoulos-Borrill et al., 2020; Feldmann and Levermann, 2015; Reese et al., 2020). However, these studies use ocean melting parametrisations, that contain simplifications of important ocean physics such as, Coriolis force, ocean mixing parameterisations, barotropic ocean flow, and lateral variation in the direction parallel to the

grounding line and hence lack spatial variation in melt rates caused by differences in ocean velocity and temperature. Therefore, these models incompletely represent the complex interactions between ice shelf geometry, ocean dynamics and melt rates, potentially leading to overestimations in rates of grounding line retreat and mass loss (Seroussi et al., 2017; De Rydt and Gudmundsson, 2016). To accurately simulate ice evolution, a coupled ice/ocean model must be used.

Previous coupled modelling studies have used different approaches to the coupling using either an 'asynchronous' (e.g., Seroussi et al., 2017; De Rydt and Gudmundsson, 2016; Naughten et al., 2021) or a 'synchronous' coupling (e.g., Goldberg and Holland, 2022; Goldberg et al., 2018; Jordan et al., 2018). Synchronous coupling involves continuously changing the ice geometry during the ocean simulation at the ocean model timestep, while asynchronous coupling involves information being exchanged every one or few ice model timesteps, with the ice geometry and ocean state modified at each coupling period.

Regional coupled models have been used to simulate parts of the Amundsen Sea sector, for example simulating Thwaites Glacier over 50 years (Seroussi et al., 2017), and the Pope, Smith and Kohler glaciers (Goldberg and Holland, 2022).

In this study we use a new synchronously coupled ice/ocean model of the Amundsen Sea sector to simulate the evolution of its ice streams over the next 180 years. We consider three different idealised forcing scenarios: no basal melting, and cold and warm Amundsen Sea conditions. We use these simulations to explore both the future evolution of the ice sheet in this sector, and the physical processes that determine the speed of the retreat. This allows us to better understand the mechanisms by which the ice loss is sensitive to future ocean conditions, and thus to anthropogenic forcing.

## 2 Methods

### 2.1 Ice model

In the coupled ice/ocean model, ice velocities are calculated using the Wavelet-based, Adaptive-grid, Vertically Integrated ice sheet model, WAVI (Arthern et al., 2015; Arthern and Williams, 2017). WAVI is a finite volume ice sheet model including a treatment of both membrane and simplified vertical shear stresses as described by Goldberg (2011). We use a numerical model with a time step of 20 days and a 2 km horizontal resolution covering the whole Amundsen Sea sector domain shown in Figure 1a. This horizontal resolution is found to be appropriate in a recent study, currently under review, testing the impact of resolution on grounding line retreat and SLR contributions, with a nearly identical configuration of the Amundsen Sea sector (Williams et al., PREPRINT) (https://doi.org/10.21203/rs.3.rs-3405435/v1). The ice rheology is described using Glen's flow law, with an exponent of n = 3. A Weertman' sliding law is used (Weertman, 1964), for which basal sliding drag scales with the cube root of sliding velocity, multiplied by a basal sliding drag coefficient C. As described in Arthern et al. (2015), WAVI uses a data assimilation method to match modelled ice velocities and rates of thickness changes with observations of the MEaSUREs 2014/15 surface velocities (Mouginot et al., 2017a; Mouginot et al., 2017b) and rate of change of surface elevation from Smith et al. (2020), resulting in an initial state representing conditions in approximately 2015. Partially grounded cells are utilised using a sub-grid parametrization to better represent the grounding line (e.g., Arthern and Williams, 2017; Pattyn et al., 2006; Cornford et al., 2012; Seroussi et al., 2014), where the grounding fraction is used to proportionally apply the Weertman sliding drag coefficient. An ice model relaxation is then run for a set period of time (4000 years). During this relaxation the grounding line and the thickness of ice shelves remain fixed, but the grounded ice thickness is allowed to change (see Arthern et al. 2015 for full details). This brings the flux divergence into much better agreement with observations of accumulation and rates of ice thickness change but at the cost of the surface elevation and ice velocities agreeing less well with observations (see Appendix A). Ice thickness prior to relaxation and bathymetry fields are from BedmachineV3 (Morlighem et al., 2020; Morlighem, 2022), where a minimum thickness of 50 m is applied to determine the initial ice extent, which includes the current gaps between the east and west of Thwaites Ice Shelf,. The ice front and outer catchment boundaries are kept fixed throughout the simulations. Accumulation and englacial temperatures are kept constant in the forward simulations, using data sets from Arthern et al. (2006) and Pattyn (2010) respectively.

## 2.2 Ocean model

We use the Massachusetts Institute of Technology general circulation model (MITgcm; (Marshall et al., 1997)) to simulate the ocean circulation in the Amundsen Sea. The model grid uses polar stereographic Cartesian coordinates with horizontal resolution of 2 km and vertical resolution of 20 m, and we use a time step of 200 seconds. The ocean model domain extent is shown by the black box in Figure 1a. Note that this does not include the whole continental shelf, so we apply ocean boundary conditions and velocities to the northern and western open boundaries of the domain to impose prescribed Amundsen Sea conditions in idealised 'warm' and 'cold' experiments. We also examine a third 'no melting' case in which no ocean-driven melting is applied to the ice shelves. This study focuses solely on the ice sheet, ice shelf and ocean interactions driven by wider Amundsen Sea conditions, so sea ice and other freshwater sources/sinks are not included in the simulations, and no atmospheric forcing is applied over the model domain.

The boundary conditions in warm and cold experiments are as follows. Following previous studies, an idealised, piecewise linear, vertical profile is applied to replicate the warm CDW layer below cold Winter Water, with a thermocline 400 m thick (De Rydt et al., 2014). The base of the thermocline is placed at 600 m depth in the warm scenario and lowered to 800 m depth in the cold scenario (Figure 1c, d). These profiles correspond to the warmest and coldest observed Amundsen Sea conditions, in 2009 and 2012 respectively (Dutrieux et al., 2014). On the northern boundary the warm CDW layer has a temperature of 1.2°C and a salinity of 34.7 (PSU), whereas the western boundary is forced with a more modified CDW layer, with temperature 0.6°C and salinity 34.6 (PSU) (Figure 1c, d). In all simulations, at the boundaries, we apply average ocean velocities (1965-2015) from a larger regional model (Holland et al., 2022; Naughten et al., 2022), and we restore sea level to zero.

Ocean only simulations, with fixed initial ice geometry, are simulated for 2 years, as a spin up before coupling. The resultant conditions in the ocean model are designed to match the spatial distribution of present day observed maximum subsurface temperatures over the region (Dutrieux et al., 2014), as shown in Figure 2a,b. The warm and cold oceanic forcing cases enable different amounts of CDW to reach the base of the present day Thwaites and Pine Island Glacier (PIG) ice shelves (Figure 2e, f, i, j). The initial melt rates, after the 2 year ocean only spin-up, of the Thwaites and PIG ice shelves for the warm and cold simulations are shown in Figure 2c, d, g, h.

Ice shelf melting is represented by a standard three-equation formulation (Holland and Jenkins, 1999). We tune the dimensionless ice shelf melting drag coefficient in this parametrization to 0.008. This drag coefficient parameterises the ocean stress on the ice base as a function of the ocean model's mixed layer velocities, in order to calculate turbulent ocean heat and salt fluxes for use in the melting calculation(Jenkins et al., 2010). For PIG and Thwaites ice shelves combined, this value produces the closest average match between the initial MITgcm ice shelf melt rates and those that are implicit in the WAVI initialisation (see Appendix B). This minimises the 'coupling shock' - the response of the ice model to any mismatch between

these two fields -, which occurs when the coupled model simulation commences. Without this calibration, the ice sheet trajectory could be impacted, potentially for many decades (Goldberg and Holland, 2022), by the adjustment of the ice due to the transition from implicit initialised melt rates to arbitrarily different ocean model melt rates at the start of the simulation. Note also that this tuned value is close to a value of 0.01 derived from observations (Jenkins et al., 2010).

Bedmachine version 3 is used as the basis for the bed/seabed geometry throughout the model. However, we found that without modification this dataset led to rapid grounding and advance of PIG. Closer inspection revealed that the seabed bathymetry may have been underestimated near the grounding line of this glacier in Bedmachine version 3 (Appendix C). Therefore, throughout the domain we deepen the seabed wherever needed to achieve a minimum water column thickness of 280 m on the staggered Arakawa C-grid's velocity grid points, located on the grid faces, tapering towards grounded ice over 6 km down to a minimum water column thickness of 140 m. This procedure is only applied to cells with no ice basal sliding drag in the initial state and is only done once at the start of the simulation before the WAVI ice sheet model is initialized and relaxed, rather than being an ongoing process. This edited bathymetry is used in both MITgcm and WAVI.

## 2.3 Ice/Ocean coupling

This study uses a new, synchronously coupled ice/ocean model, which combines the WAVI ice sheet model with the MITgcm. Crucially, this coupling occurs through the MITgcm ice sheet package STREAMICE (Goldberg and Heimbach, 2013), making use of previous synchronous coupling developments (Jordan et al., 2018; Goldberg et al., 2018), which enables the STREAMICE ice thickness to evolve continuously in the ocean model. The basic concept of the new model is that WAVI solves for the ice velocities, MITgcm STREAMICE solves for the ice thickness, and the MITgcm ocean model solves for all ocean properties, including melting and is also where the ice thickness evolves. Dividing up computations in this way allows for a fully synchronous coupling because both the ice thickness and ocean properties are solved within the MITgcm on the ocean timestep. This has two advantages: firstly, the ice thickness and ocean free surface equations are solved simultaneously, allowing full conservation of mass in the ice and ocean coupled system. Secondly, the melt rate responds instantly to changes in ice thickness.

In principle, this approach is no more expensive than other coupling approaches because the two-dimensional ice thickness equations can be solved on the ocean timestep with negligible computational expense compared to that incurred in solving three-dimensional ocean equations. However, the approach does require the MITgcm grid to exist wherever ice may go afloat during the simulation, since grounding-line retreat in MITgcm is accomplished naturally. MITgcm solves the ocean free surface equation every ocean timestep in order to conserve mass, which naturally inflates the water column in the MITgcm ocean model wherever the pressure loading of the ice decreases below floatation (Goldberg et al., 2018). In regions of ice which are not floating, a thin subglacial layer is specified in order to enable the expansion of the ocean column during

grounding line retreat (Goldberg et al., 2018; Jordan et al., 2018). We set it to be 4 m thick but could have been set to any relatively small thickness compared to the ocean model vertical resolution. This small value has been previously demonstrated to have no impact on the evolution of the coupled system (Goldberg et al., 2018). This layer, which has no effect on basal sliding drag computed by the ice model, is treated as a porous medium, with Darcy flow used to connect the subglacial cells to each other (Goldberg et al., 2018).

Using an MITgcm domain covering the entire catchment of the Amundsen Sea sector would be very inefficient, since it would include large areas of grounded ice that would never go afloat in the ocean/STREAMICE grid. To avoid this, we introduce the concept of 'ice-only' and 'coupled' domains (Figure 1a): In the ice-only domain, WAVI is used to solve for both ice sheet velocities and thickness, while in the coupled domain, WAVI solves for ice velocities and MITgcm solves for ice thickness and ocean properties, as described above. The coupled domain only needs to extend far enough inland to accommodate the grounding-line retreat occurring during a projection, which for this study was determined using test simulations.

The coupled model solution procedure is split into coupling periods, chosen to equal the WAVI timestep of 20 days, meaning that the WAVI model state is fixed during each coupling period. Over each coupling period (Figure 1b) the following takes place: (1) WAVI calculates ice velocities over the whole ice domain for one 20-day timestep; (2) WAVI steps forward the ice thickness in the ice-only domain for one 20-day time step; (3) MITgcm receives the ice velocity for the coupled domain and the updated ice thickness on its boundaries, and sub-cycles the ice thickness, grounding line and ocean properties within the coupled domain using the 200 s ocean timestep for the full 20 day coupling period; (4) MITgcm passes the new ice thickness and therefore the new ice grounding line for the coupled domain back to WAVI and the next coupling period commences. Therefore, the choice of the length of the coupling period determines the fastest response time for which the ice velocities can respond to changes in buttressing in the coupled domain, where the ice thickness changes with the ocean timestep. The WAVI boundary cells outside of the coupled domain are passed to the MITgcm domain and held fixed while MITgcm runs, and then the thickness throughout the coupled domain is passed from MITgcm to WAVI after each coupling step. This procedure, in addition to WAVI being fixed during each coupling period, keeps the two models/domains ice thicknesses from diverging and ensures a smooth transition between the two domains.

For consistency, the same advection scheme that is used in WAVI (Arthern et al., 2015), has been coded into STREAMICE. Ice divergences are updated with the changing ice thickness in STREAMICE every ocean timestep, while the ice velocity remains fixed over each coupling period. We adopt the principle that the ocean may only melt ice that has no basal sliding drag applied (Arthern and Williams, 2017), so the WAVI ice basal sliding drag field is passed to MITgcm to decide where melting can occur during each coupling period. This means that no melting can occur on partially grounded cells, as recommend by Seroussi and Morlighem (2018). In addition, this means that if a cell becomes fully ungrounded in MITgcm,

195  ice shelf melting only occurs once the coupling period finishes and a new one starts, which updates the ice thickness in WAVI, and subsequently passes back a new basal sliding drag field. However, if, during a coupling period, a grid cell becomes grounded in MITgcm, melting on this cell is immediately switched off. In both the WAVI and STREAMICE models a minimum ice thickness of 50 m is applied.

200

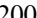

**FIGURE 1:** (a) Outline of the whole Amundsen Sea sector domain. The black box shows the outline of the coupled domain; the ice only domain is defined as everywhere outside of this box. The coloured contours represent the three regions used in analysis: PIG (purple), Thwaites (red), and Smith (cyan). These areas are edited by hand from the Zwally et al. (2012) basins. (b) A schematic diagram of the three
205  grouped steps of the coupled model applied during one coupling period, which are repeated during the coupled simulations. (c) Potential temperature boundary conditions for the Northern (solid line) and Western (dashed line) boundaries for the warm (red) and cold (blue) cases. (d) as in (c) but for salinity.

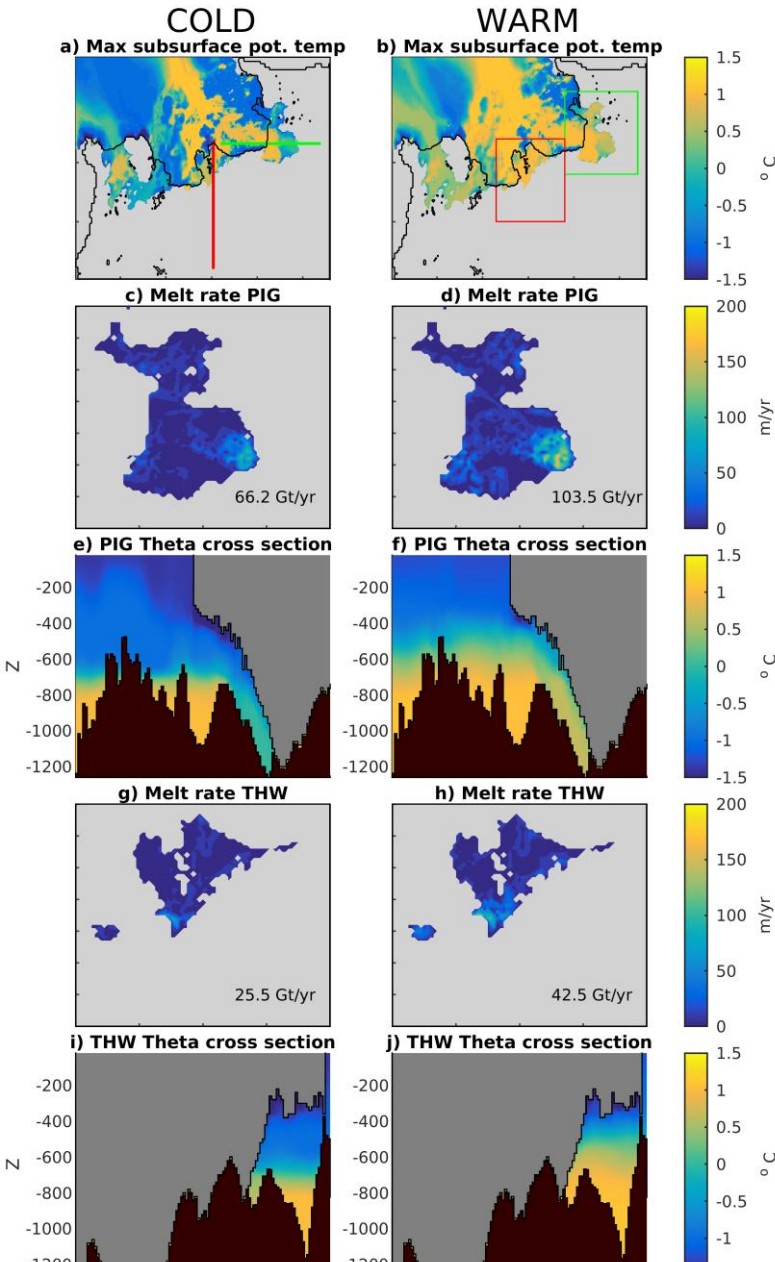

**FIGURE 2:** (a) Initial maximum subsurface potential temperature over the ice/ocean domain for cold case**.** (c) Initial melt rate under PIG Ice Shelf in the cold forcing case over the green box in (b). The label shows the total initial meltwater flux. (e) Cross-section through PIG Ice Shelf taken along the west-east green line in (a) showing initial potential temperature in the cold case. (g) Initial melt rate over Thwaites Ice Shelf in the cold forcing case over the red box in (b). (i) Cross-section through Thwaites Ice Shelf taken along the south-north red line in (a), showing initial potential temperature in the cold case. (b), (d), (f), (h), (j), as in (a), (c), (e), (g), (i) respectively, but for the warm forcing case.

# 3 Results

## 3.1 Simulated evolution of the Amundsen Sea sector

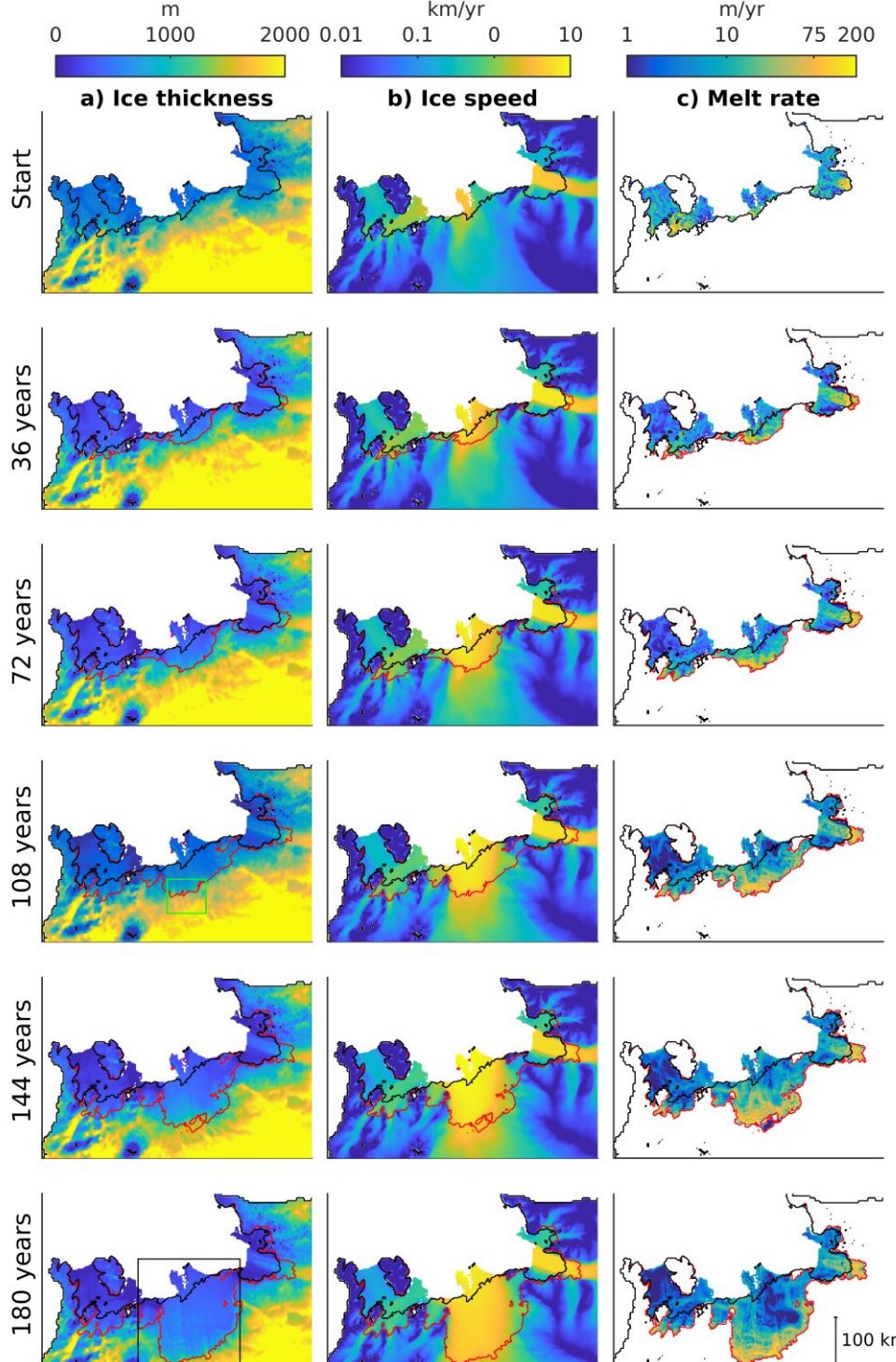

220

**FIGURE 3:** Warm forcing case evolution over 180 years. Panels show snapshots of (a) ice thickness, (b) ice speed and (c) melt rates plotted on a logarithmic scale, taken every 36 years throughout the simulation. Initial (black) and final (red) grounding lines are included.

Following initialisation, the coupled ice/ocean model is run forward for 180 years under both forcing cases, as well as a zero-225 melting case. The evolution of the warm simulation is shown in Figure 3. The most prominent feature of the simulation is the acceleration (figure 3b) and grounding-line retreat of Thwaites Glacier (figure 3a). At the start of the simulation, Thwaites Ice Shelf hosts the observed fast-flowing western shelf and slower eastern ice shelf (figure 3b), where a pinning point restrains the flow of ice. However, as the simulation progresses, both sides of the ice shelf accelerate, reaching speeds of up to ~10 km/yr, but then the whole ice shelf shows signs of deceleration between 144 years and 180 years. In addition, Thwaites Glacier 230 experiences substantial grounding-line retreat during this simulation, approximately 130 km of north-south retreat, leading to a reduction in the total mass of grounded ice and the formation of new larger ice shelf cavity, which features high melt rates near the grounding line (figure 3c). The fixed ice front leads to a large and thin future Thwaites Ice Shelf, which may lead to artificially elevated melting overall, though only low ice shelf melt rates occur on the thinner ice (figure 3a,c). However, such trends in ice speed and grounding line retreat are restricted to Thwaites: neither PIG nor Smith Glacier experiences the same 235 level of acceleration or retreat, although we do observe an initial acceleration of the PIG (which is later reversed), and some grounding line retreat in both PIG and Smith.

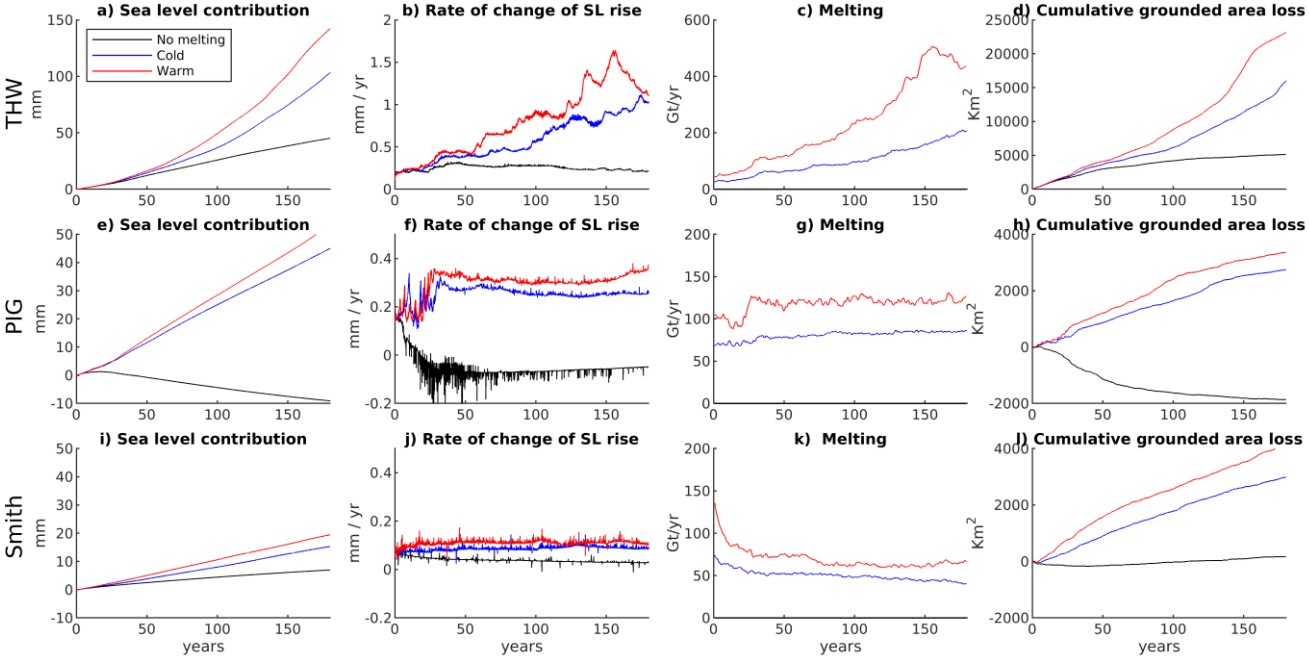

**FIGURE 4:** Timeseries shown for the three cases no melt (black line), cold (blue line), and warm (red line). (a) Cumulative global sea level contribution from the Thwaites area. (b) Rate of change of global sea level contribution for Thwaites area. (c) Total ice shelf melt rate from

the Thwaites area, with a 2-year running average applied. (d) Cumulative grounded area loss from the Thwaites area. (e - h) Same as (a - d) but for the PIG area. (i - l) Same as (a - d) but for the Smith area.

In order to analyse SLR contributions, melting and grounded ice area loss from different regions, and to compare the three forcing cases, we divide the domain into three areas: Thwaites, PIG and Smith (Figure 1a). These areas are edited from the (Zwally et al., 2012) basins, where the Smith area has been separated out. In addition, the boundary between the PIG and Thwaites areas is edited by hand to address the otherwise mis-assignment of SLR contributions, melting and grounding ice loss to PIG, as Thwaites retreats (Fig 3).The model projects notable SLR contributions from all three regions, even in the zero-melting case for the Thwaites and Smith areas, which has important implications. This result demonstrates that the initialised ice-sheet state in WAVI is intrinsically unsteady and will continue the present-day ice loss for some period of time irrespective of future climatic forcing. This implies that a tipping point may have been passed at some point for this particular initialised ice sheet, likely during the 20[th] century (Mouginot et al., 2014), and some amount of SLR (at least ~40 mm in our simulations) is now committed. Zero melting can never occur in the real world, because even seawater at the surface freezing point will drive melting at depth due to the pressure decrease in the freezing temperature, so this is a very conservative test of the presence of committed ice change. In the PIG area there is a negative SLR contribution for the zero-melting case (mass gain), though the contribution from PIG is dependent on the particular choice of bathymetry deepening that is implemented (Appendix C). Notably, we observe the largest zero-melting SLR contribution from the Thwaites area, leading to ~45 mm in 180 years, with a near constant SLR rate. For the Thwaites area we also obtain some grounding line retreat, though the rate of loss reduces during the simulation, suggesting that the SLR contribution would stabilise after a further period of simulation.

The cold and warm cases are far more realistic since they are based on observed Amundsen Sea conditions. Both scenarios contribute much more SLR than the zero-melting case (figure 4a, e, i). In both cases, the Thwaites area dominates SLR contributions from the sector, providing approximately two-thirds of the total, and responds differently to melting than the PIG and Smith areas. In PIG and Smith areas, the rates of SLR with melting are approximately constant but are higher than the rates of SLR in the zero-melting case, increasing to ~0.3 mm/yr and ~0.1 mm/yr in the warm case, respectively, (figure 4f, j). However, in the Thwaites area with melting a trend emerges in the SLR rate (figure 4b), increasing from ~0.2 mm/yr at the start of the simulation to ~1.1 mm/yr at the end, in the warm case. In addition, we observe rapid jumps in the SLR rate for Thwaites that are not present in the other areas. The increase in the SLR rate in the Thwaites results in an approximately quadratic SLR contribution (figure 4a), rather than the more linear SLR contributions obtained for PIG and Smith. In all three areas 'noise' is present in the SLR rates, due to the instant effects of the changing basal sliding drag field, though this is harder to see for the Thwaites area due to the larger y-axis scale. In addition, we observe an increase in the total melting that occurs from the Thwaites Ice Shelf as the simulations progresses: over the 180 years of the simulation, the total melt flux from Thwaites increases by an order of magnitude, from ~45 (~25, respectively) Gt/yr at the start of the simulation to over ~440 (~210) Gt/yr at the end in the warm (cold) case (figure 4c), which occurs despite constant oceanic boundary forcings during

the simulation. This is explored further in Section 3.3. In the PIG and Smith areas, the total melt flux is approximately constant (figure 4g, k). However, we do observe a strong correlation between the cumulative integral of melting and the SLR contribution individually for each region (Appendix D). In addition, while the effect of the inclusion is small, we do include ice shelf melting that occurs on ice that has reached the minimum thickness, as while this melting is not allowed to thin the ice further it still has a partial glaciological effect, by stopping the ice from thickening. Furthermore, this melting still has an oceanographic effect in the simulation by cooling and freshening the ocean.

For all regions, we obtain consistently higher SLR rates and grounded ice area loss in the warm forcing case compared to the cold forcing case (figure 4b, f, j). However, as a fraction of the total SLR in these models, the difference between warm and cold scenarios is remarkably small, only ~27% of total SLR from all regions, by the end of the simulations. Given that these scenarios bracket the coldest and warmest ocean conditions on record in the Amundsen Sea, this suggests that the future SLR from this region is only weakly influenced by variations within the observed range of present day ocean conditions: while melting is important to the SLR, its typical climatic variations are less so. Despite this, in the Thwaites area we observe different timings and strengths in the jumps in the SLR rate between the cold and warm cases (figure 4b). We also observe an increasing difference between the cold and warm total melt rates for Thwaites Ice Shelf (figure 4c), while for the PIG and Smith areas this difference remains approximately constant (figure 4g, k). Additionally, there is a larger relative difference between the melting and no melting cases for the PIG area compared to Thwaites. In the PIG area, ice shelf melting prevents the ice shelf from thickening and re-grounding on the bathymetric ridge below it. In the no melt case, the ice shelf regrounds on this ridge; the buttressing provided by this re-grounding leads to the large differences between the no melt and melting cases for the PIG area (though this is dependent on bathymetry deepening). While there is a smaller relative difference for the Thwaites area, the presence of melting still has a significant impact, changing the SLR contribution from linearly increasing to quadratic.

Overall, we conclude that melting and SLR have a fundamentally different response in the Thwaites area than the other areas, leading to an increasing SLR rate and total melting rate. In addition, the Thwaites area dominates the SLR contribution for the Amundsen Sea sector over the 180 years simulated. Therefore, the remainder of this study focusses upon the processes underlying this behaviour in the Thwaites area.

**3.2 Thwaites Glacier retreat and ice shelf pinning points**

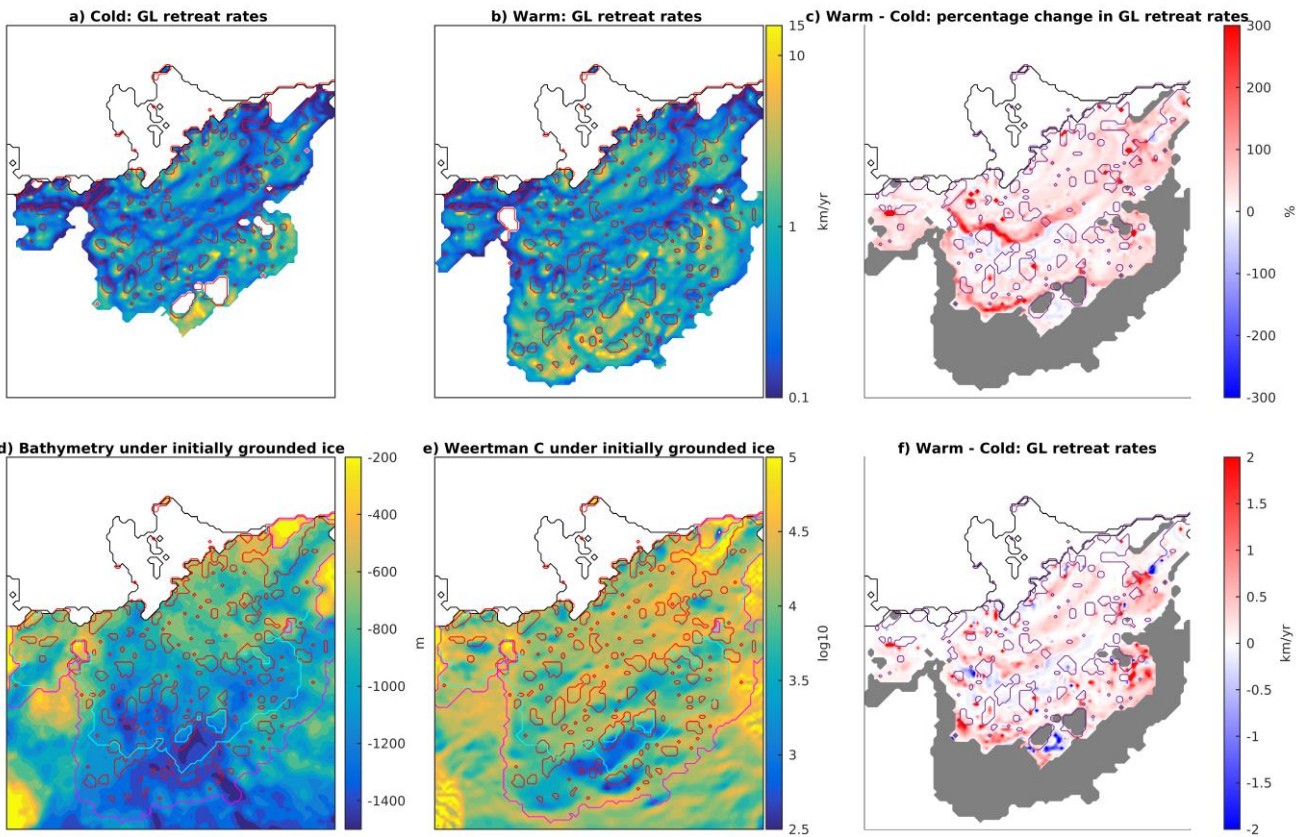

**FIGURE 5:** (a-b) Grounding line retreat rates over the 180 year simulations in the cold (a) and warm (b) indicated by colours, with colour bar shown in (b). The red contours in (a, b, d, e) and the purple contours in (c ,f) show the presence of isolated pinning points during the simulation. (c) Percentage change in grounding line retreat rates between the cold and warm cases (i.e., the percentage difference between those retreat rates shown in (a) and (b) respectively). (d-e) Bathymetry (d) and Weertman C coefficient (e) shown under initially grounded ice with final extents of the grounding line retreat in the warm (magenta) and cold (cyan) cases. (f) The difference in grounding line retreat rates between the cold and warm cases (colours) as well as final grounding line area (grey area). The area shown in each panel corresponds to that displayed as a black box in Figure 3.

Figure 5 shows quantities related to Thwaites grounding line retreat, which is one of the key features that differentiates this area from the PIG and Smith areas. Over the simulation, the warm case has a larger area of grounding-line retreat than the cold case, with a faster rate on average (figure 5 a, b, c, f). Grounding line retreat rates are calculated from the discrete migration of the grounding line across grid cells. Specifically, the retreat rate in each grid cell is calculated as $\Delta x_{GL} / \Delta t_{GL}$, where $\Delta t_{GL}$ is the time between when the subject cell first becomes a grounding line cell (any of the 8 adjacent cells are floating) to when that cell ungrounds, and $\Delta x_{GL}$ is the grid cell width or diagonal extent depending on which adjacent cell was first floating.

Figures 5d and 5e show the bed depth and 'Weertman C' drag coefficient over the ice area that is initially grounded. A lower Weertman coefficient corresponds to a more slippery bed. In addition, the bathymetry in this area generally deepens inland, which could promote grounding line retreat (Weertman, 1974; Schoof, 2007). In both warm and cold cases, highly
heterogeneous retreat rates are observed (figure 5a, b), with areas of fast retreat as high as ~10 km/y in proximity to areas of much slower retreat. Towards the end of the warm case simulation, the grounding line experiences rapid retreat (Figure 5b) across a deep and slippery bed section (Figure 5d-e) before slowing down as it encounters a ridge of shallower and less slippery bed (Figure 5d-e), where it remains until the end of the simulation. These features explain the large variations in the Thwaites area SLR rate in the last ~50 years of the simulation (Figure 4b). In particular, the retreat onto the shallower and less slippery
ridge at the end of the warm simulation decreases the ice flux across the grounding line and corresponds with the Thwaites region's decreasing SLR rate during the last 25 years of the simulation.

Comparing the cold and warm scenarios we see the effect of increased melting on grounding-line retreat rates. More specifically, we observe areas of elevated grounding line retreat rate in the northern part of the ungrounding area, below which
an area with minimal sensitivity to melting scenario, and again by increased rates in the south (Figure 5f). There are very large percentage increases in grounding-line retreat rates in two clear bands on the retreated area (Figure 5c), where two bands of slow retreats rates in the cold case are not present in the warm case.

In addition, we observe the formation of many pinning points as Thwaites Glacier retreats, shown by red and purple contours
in Figure 5 for the different cases. An ice grid cell is flagged as a 'pinning point' if it is grounded but separated from the main grounded ice sheet by floating ice. The red and purple contour is then drawn around all cells that are flagged as pinning points in any one of the outputs. The pinning points are generally located on areas of shallower bathymetry that are downstream of areas of deeper bathymetry (Figure 5d). The pinning points typically feature slower retreat rates, and inshore of these pinning points we generally observe areas of faster retreat (Figure 5a, b). These pinning points are crucial for the future evolution of
Thwaites Glacier, as they provide drag and buttressing as the ice retreats. The importance of these pinning points is determined by their size, position, and duration, so we examine these features and their relationship to SLR rates from the Thwaites area.

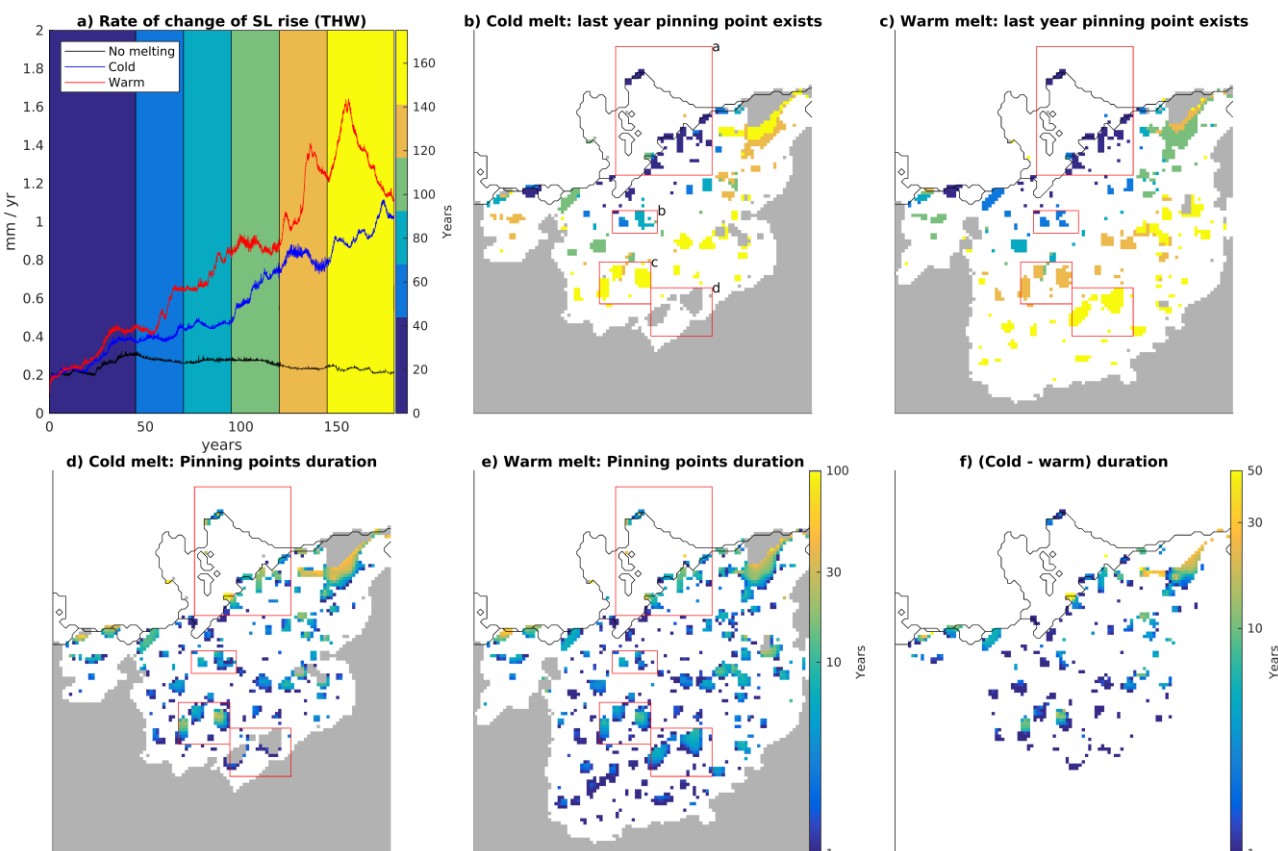


**FIGURE 6:** (a) SLR contribution rates for Thwaites area for warm (red), cold (blue) and no melt (black) cases. The background colour indicates the time in the simulation according to bands of 25 years, with a longer starting (ending) band of 45 (35) years. (b) Colours indicate the band of the final year that pinning point cells exists for cold melt case, according to the colours shown in (a). Note that only pinning points that exist for over 1 year are shown. Red labelled boxes refer to groups of pinning points discussed in the text. (c) Same as (b) but for

warm case. (d-e) Pinning point duration, taken as the time from isolation to ungrounding, for cold (d) and warm (e) cases. (f) Difference in pinning point duration between cold and warm cases (i.e. the difference the data shown in d and e), where pinning point locations match. In (b-e), grey regions indicate areas which do not become ungrounded during the simulation. The area shown in Figures b-f is shown with a black box in Figure 3.


Figure 6b-c shows the ungrounding time of pinning points in the cold and warm cases, respectively. By examining these times, we can compare how the ungrounding of pinning points relate to jumps in SLR rate that we observe from the Thwaites area. The labelled red boxes in Figure 6b show the key groups of pinning points in the simulations.

Within the first 60 years, the SLR rates of the warm and cold forced cases diverge (Figure 6a). In the warm case, pinning point group 'a' completely ungrounds between years 25 and 60 (Figure 6c), with the final ungrounding of the last pinning point in group 'a' coinciding with the ungrounding of group 'b'. The removal of the combined associate buttressing leads to a large jump in SLR rate in the warm simulation at ~60 years. However, in the cold simulation, some of these pinning points in group 'a' remain grounded until beyond 95 years (Figure 6b), leading to a period of relatively steady SLR rate in this simulation

(Figure 6a). In the zero-melting case, part of pinning point group 'a' and all of 'b' remain grounded throughout the simulation (not shown), and SLR rates remain approximately constant as a result (Figure 6a). In the warm simulation, group 'c' becomes ungrounded by ~130 years, causing a rapid increase in SLR rate, and the loss of group 'd' at ~145 years leads to another large jump. The cold simulation loses group 'c' at ~170 years, leading to a small jump in SLR rate, while group 'd' remains grounded at the end of the simulation.


     Figure 6d-e shows the duration of the pinning points- the time between separation from the main body of grounded ice to ungrounding. There is a large variation in pinning point durations, with some lasting less than a year while others persist for decades. In general, the duration of pinning points is lower in the warm case (Figure 6d-f), with an average pinning point duration of ~6 years, over matching pinning locations, compared to ~10 years in the cold case, leading to the different

ungrounding timings and SLR rates as described above. Reducing the duration of pinning points increases the intensity of periods of rapid ice acceleration and grounding line retreat, leading to the differences in the spatial map of grounding line retreat rates between the cold wand warm cases shown in Figure 5 c, f.

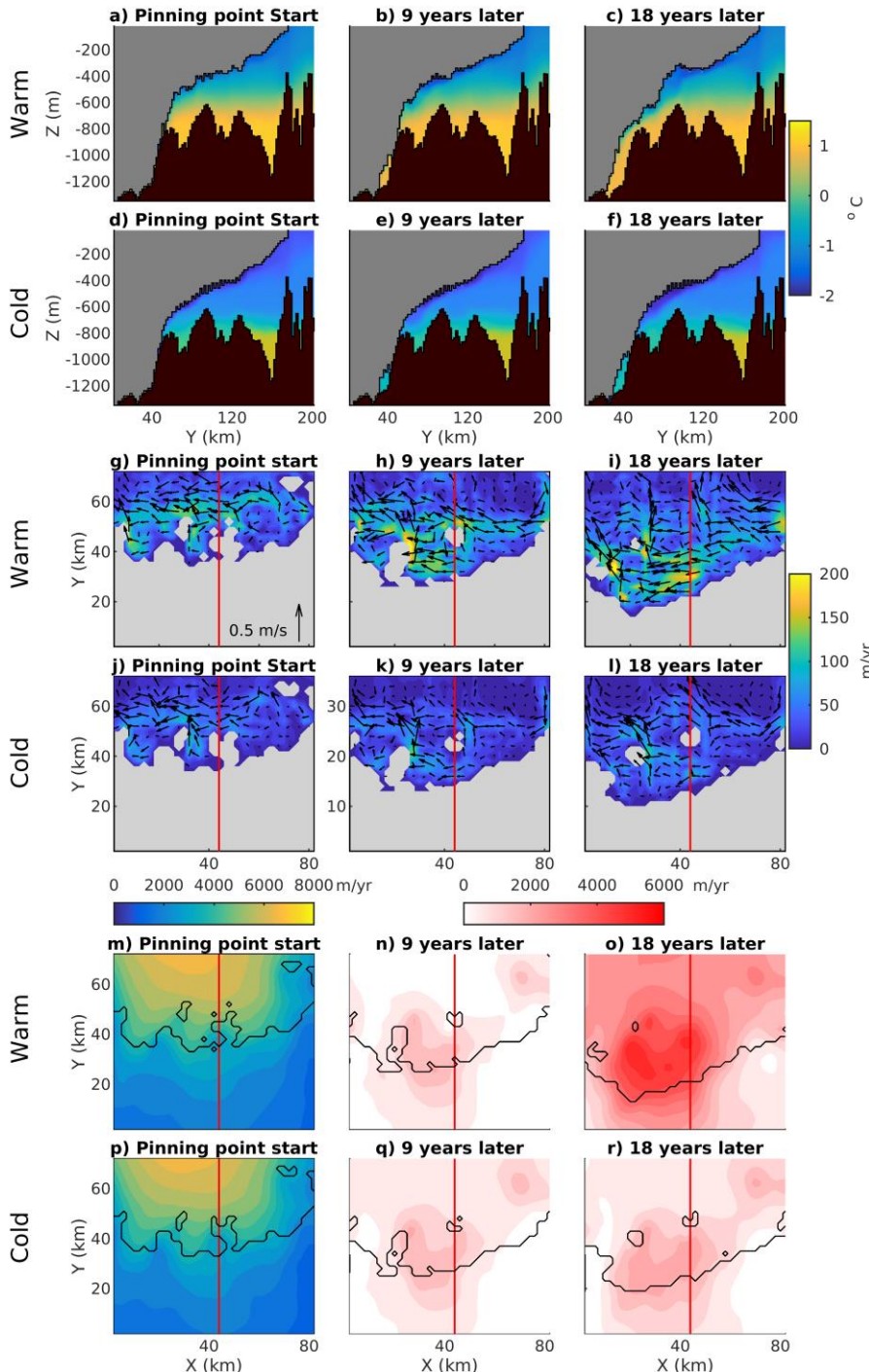


**FIGURE 7:** Evolution of conditions as grounding line retreats over pinning points. Snapshots every 9 years starting from the date of group 'c' pinning point formation in the warm and cold cases (118 and 144 years respectively). (a-c) Cross section through the Thwaites Ice Shelf, taken along the red line in (g) extended to ice front, for warm case. (d-f) Same as (a-c) but for cold case. (g-i) Melt rates for warm case, with

arrows showing ice shelf boundary layer ocean velocities (j-l) Same as (g-i) but for cold case. (m) Ice speed snapshots at start of pinning point formation and (n, o) show differences for warm case. (p-r) Same as (m-o) but for cold case. Area shown in (i-x) is shown in figure 3 as a small green box.

To illustrate the key role of pinning points, we now focus on the ungrounding of group 'c', which causes the rapid jump in SLR rate at year 130 in the warm simulation, and the smaller jump at year 170 in the cold simulation. Figure 7 shows the evolution of the two simulations, starting from their individual dates of formation of the group 'c' pinning points. Figures 7a-l show the ice geometry and ocean conditions throughout the subsequent evolution. To begin with, the grounding line is located in shallower bathymetry at the top of a retrograde slope (figure 7a, d). Ungrounding then occurs laterally around this point, including upstream, encircling the pinning point and leaving it isolated from the rest of the grounded ice (figure 7h, k). Note that the pinning point is grounded on the side of the shallower bathymetry, rather than on top of the bathymetric feature. With the grounding line now in a deeper bed an acceleration is expected, although the pinning point continues to provide basal sliding drag that resists the flow. The advection of thicker ice from the deeper bed upstream enables the ice to remain grounded on the shallower bed beneath the pinning point. Therefore, only a small acceleration is observed at 9 years, while the ice remains grounded on the pinning point (Figure 7m-r).

However, melting now occurs in the newly-opened cavity upstream of the pinning point. This thins the ice, enlarging this cavity, enabling greater oceanic connection, and leading to higher ocean velocities and melt rates (figure 7h, k). This melting and thinning feedback eventually leads to sufficient thinning upstream of the pinning point for it to unground completely (Figure 7i). The resulting loss in buttressing leads to a large increase in ice speed (figure 7o), further rapid grounding line retreat and a jump in SLR rate. Therefore, the distribution and strength of localised melt rate patterns strongly determines the duration of these pinning points and thus the overall ice retreat. This is shown clearly in Figure 7g-l, with the warm case leading to higher melt rates and a much faster ungrounding of the pinning point. The melt rates around these pinning points are highly heterogeneous (Figure 7g-l), with elevated melt rates typically occurring on their eastern side, where rapid ocean currents drive high melt rates. These rapid currents occur where buoyancy-driven meltwater flow is trapped against the grounding line by Coriolis force (Holland and Feltham, 2006).

### 3.3 Thwaites Ice Shelf geometric melting trends

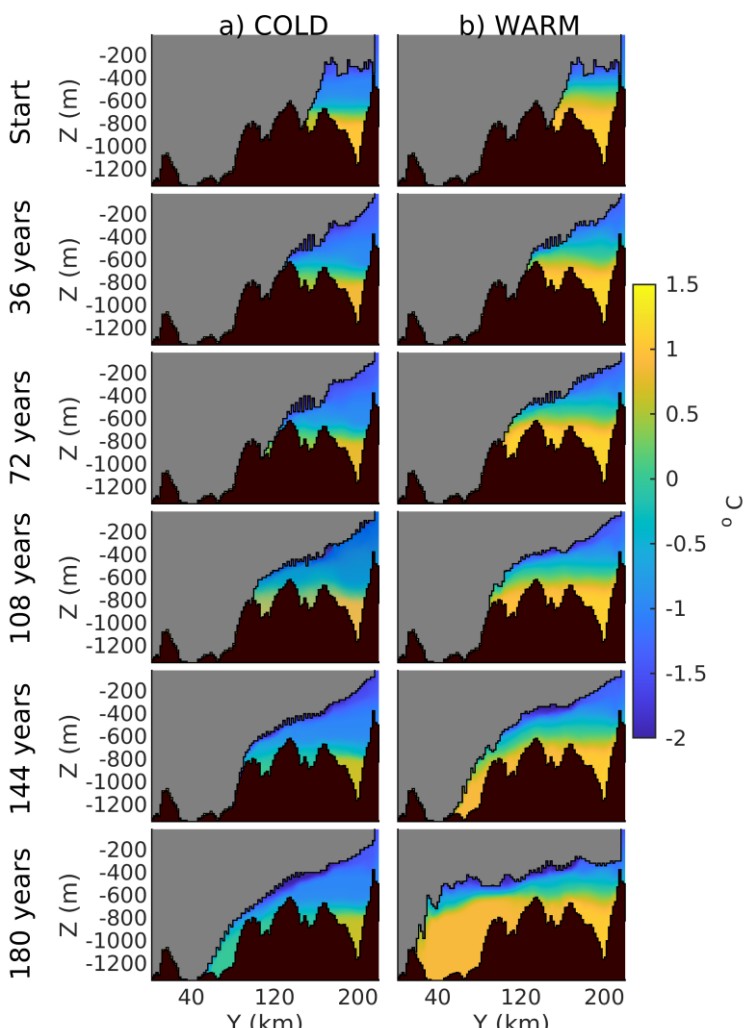

**FIGURE 8:** Cross sections of potential temperature beneath Thwaites Ice Shelf for cold (left) and warm (right) cases, taken every 36 years, along the same section as shown in Figure 7 and Figure 2i, j, which is, represented in Figure 2a as the red line.

An increasing trend in the total ice shelf melting occurs over the evolving Thwaites Ice Shelf, during both the warm and cold forcing simulations (Figure 4c). The wider oceanic forcings are fixed, so these trends must be driven by geometric changes in the ice shelf cavity. In Figure 8 we show how the Thwaites cavity geometry and ocean conditions evolve in the two cases. The warm case has a higher thermocline, so the warm CDW is more easily able to flow over ridges and flood the new cavity areas. In the cold case, the access of the warmest CDW is blocked by seabed highs, with only more heavily-modified CDW reaching

the ice base. However, sufficiently warm water is still able to drive melting close to the grounding line. In both cases the ice base near the grounding line remains steeply sloped throughout the retreat.

Figure 9a shows the evolution of total melt flux from the Thwaites Ice Shelf in the warm case, for both the entire ice shelf and for ice below 600 m depth only, which is the thermocline depth in this case. For the majority of the simulation, the total melt
flux from the entire ice shelf increases, but it decreases during the last 25 years. Most of the trend in ice shelf melting occurs in the deeper ice, with melting below 600 m peaking at an increase of ~30 times its initial value, which suggests that as the grounding line retreats, an increase in ice shelf base area below the thermocline controls total melting (Figure 9b). This confirms that the trend in melting does not result from the increasing ice shelf area associated with an artificially fixed ice front. The increase in deep ice area occurs because the groundling line retreats into deeper bathymetry, but also because the
slope of the ice shelf base, below 600 m, gets shallower during the simulation, as shown for example in Figure 8. Without an increasing trend in melt flux, the thicker ice advected across the grounding line as the ice sheet retreats into deeper bathymetry would result in re-grounding on pinning points further downstream. However, towards the end of the simulation the groundling line retreats slowly onto a shallower ridge (Figure 5d), where the shallowing grounding line depth decreases the ice shelf area below 600 m, and subsequently decreases the total amount of melting below 600 m. The close correspondence in the increase
between total melting (10 to 197 Gt/yr) and ice base area (180 to 3560 km$^2$) implies that the average melt rate (m/yr) beneath deep ice, below 600 m is approximately constant at ~45 m/yr (Figure 9c). However, we do find some temporal variability in this average melt rate, with it varying up to 50 % and importantly the local melt rates are highly spatially variable, with a spatial standard deviation close to the mean value. This spatial variability is important for the effect of melting on small scale pinning points, as described above.


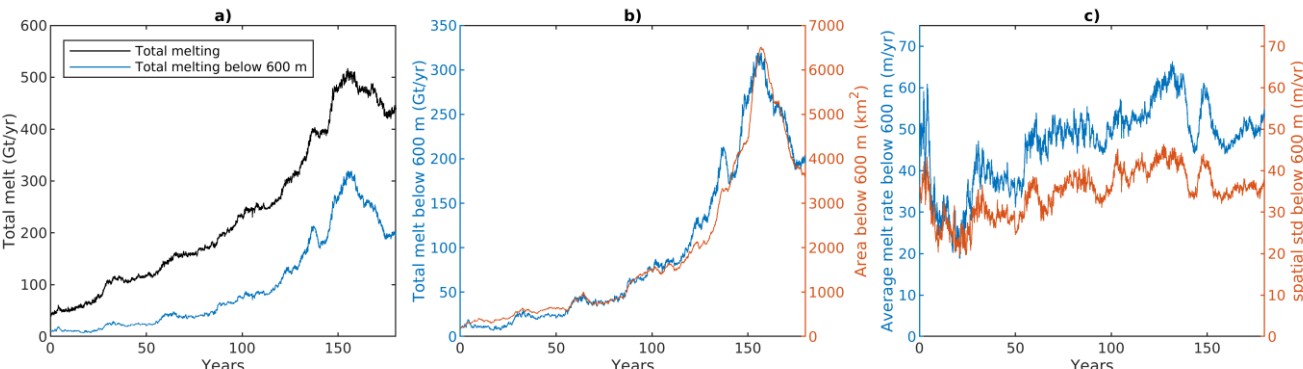

**FIGURE 9:** (a) Total melt flux from the full Thwaites Ice Shelf area (black) and from grid cells located below 600 m depth only (blue). (b) Total melt flux from grid cells below 600 m and total ice area below 600 m. (c) Average melt rate on grid cells located below 600 m depth and spatial standard deviation of these melt rate values.


## 4 Discussion

This study considers the future evolution of ice streams in the Amundsen Sea sector using a coupled ice/ocean model. Under both warm and cold forcing we find large grounding line retreat and ice acceleration from Thwaites Glacier, with the Thwaites area dominating future SLR contributions from the region. The SLR from the Thwaites area is consistent with a previous coupled model study (Seroussi et al., 2017), which considered a shorter time span of 50 years. In this study however, we show a larger sensitivity to a realistic range of ocean forcings, in the shorter and longer term, and an increasing SLR rate, which continues over the first 160 years, from the Thwaites area. This was not observed in previous coupled modelling of Thwaites Glacier (Seroussi et al., 2017), although it was observed in some ice-only simulations, depending on uncertainty in ice dynamics (Nias et al., 2019) and uncertainty in melt rates (Arthern and Williams, 2017). A previous ice-only study has set 1 mm/yr SLR rate to be the threshold that implies rapid retreat and collapse of Thwaites glacier (Joughin et al., 2014), and this is exceeded sooner in our simulations, within 125 years, than in the simulations of this previous study. However, our simulations are not long enough to see how the full implications play out over multiple centuries.

The SLR contributions in this study of 70 - 89 mm after 100 years for the Amundsen Sea sector are within the uncertainties of previous ensemble studies (Nias et al., 2019; Edwards et al., 2021). However, the SLR and stability suggestions in this study do differ from recent studies, which found that the present-day geometry is not inherently unstable when starting from a stable starting position (Hill et al., 2023), and that the Amundsen Sea sector has not tipped yet (Reese et al., 2023). We suggest that these differences arise primarily through the different ice sheet model initialisation strategies adopted, which variously use a spin up period (Reese et al., 2023), data assimilation of ice velocities into a steady state (Hill et al., 2023), or assimilation of ice velocities and observations of unsteady thinning (present study). Also, the date of initialisation and differences in resolutions, datasets used and model physics may also play an important role. Larger coupled-model ensembles are needed to assess these aspects. Without these model ensembles there is high uncertainty, for example, in SLR contributions and the timing of when pinning points become ungrounded. This study is designed to provide a small number of physically-advanced coupled simulations focusing on ice/ocean processes, rather than providing a larger but uncoupled set of predictions of future SLR contributions from the region.

In our simulations, the increase in SLR rate from the Thwaites area is caused by ocean-driven melting, and the magnitude of this increase is sensitive to different rates of melting in warm and cold scenarios. The SLR rate is governed by a balance between retreat of the grounding line into deeper bed regions, and the formation and duration of pinning points during this retreat. Crucially, this study shows the importance of ocean-driven melting in ungrounding these pinning points, reducing ice-shelf buttressing and enhancing grounding-line retreat. It should be noted that the modelled rates of ungrounding upstream of pinning points are high enough to explain recent observations of grounding-line retreat rates (Graham et al., 2022). Other studies have hypothesised about the mechanics and importance of pinning points (Thomas, 1979), and have shown the effect

of pinning points in an idealised ice/ocean coupled model (De Rydt and Gudmundsson, 2016). In this study we clearly show
the importance of these mechanisms in a synchronous coupled model of the future Thwaites Ice Shelf, with strong spatial
variability in ice shelf melting determining the duration of these pinning points and therefore their influence on SLR. Our use
of a synchronously coupled model means variations in the ice thickness are instantly felt in the ocean model's melting
calculation. This could impact the speed of the evolution of these pinning points, though further work is required to determine
the impact of this coupling on the ice dynamics. The importance of future pinning points suggests the need for more accurate
knowledge of bathymetry and bed properties in the grounded portion of Thwaites Glacier, as well as highlighting the
importance of accurately modelling the effects of pinning points in both ice sheet and ocean models. We have shown that high
resolution coupled ice-ocean models are required to investigate the effect of pinning points on ice dynamics, as these small
features need to be resolved in the ice model, and the strong spatial variability in ice shelf melting around them needs to be
recreated.

One important question raised by our study is why Thwaites Glacier appears to behave differently from the other glaciers in
the region. We speculate that this is caused by the wide trunk of this glacier. As the grounding line of the Thwaites Glacier
retreats it forms an extremely short and wide ice shelf, unlike the other glaciers in the region, whose ice shelves are confined
within embayments. As a result, the buttressing provided by lateral ice shelf margins is very weak for Thwaites Glacier; this
also explains the high sensitivity to pinning points which therefore provide the majority of the buttressing of Thwaites.
Therefore, while a recent study found that Thwaites Ice Shelf provides limited buttressing in the present day (Gudmundsson
et al., 2023), we find that future configurations of Thwaites Ice Shelf will provide important buttressing, via future pinning
points, as its groundling line retreats. The dependence of this future buttressing on ocean forcings will determine the future
SLR from this sector.

The simulations show how the geometric changes in the Thwaites Ice Shelf and its cavity can lead to an increasing ice area
being exposed to deep warm waters, leading to an increasing trend in total melt flux. Although this general depth dependence
on melting is often captured by parametrizations of melting (Asay-Davis et al., 2017), we have shown that there is strong
spatial variability in melting, and this spatial variability has important consequences for pinning point duration. Therefore, as
parametrizations of basal melting generally lack a strong physical basis and perform poorly in spatial detail (Burgard et al.,
2022), they may be expected to struggle to capture this spatial variability, affecting the evolution of these pinning points and
the resultant SLR contribution rate from Thwaites. However, future work is required, running sets of ice only simulations, to
perform an extensive comparison of the wide range of basal melt rate parametrizations. Observations of ice shelf melt rates
around such features with sufficient spatial coverage are currently lacking but are essential for improving future modelling
efforts. The increasing trend in total melt rates is needed to maintain the ice retreat, as it counteracts increased ice thickness
advection across the grounding line, which would otherwise ground the ice on bathymetry further downstream.

One of the limitations of this study is that steady, idealized, ocean forcings are applied to the northern and western boundaries. However, ocean conditions in the region are known to have strong decadal variability (Jenkins et al., 2018; Dutrieux et al., 2014). Some of the pinning points only exist for ~10 years, so decadal variability in oceanic forcings may have an important role in how quickly these pinning points are ungrounded. In addition, a superimposed anthropogenic warming trend in ocean forcing may be expected in the Amundsen Sea (Holland et al., 2022; Naughten et al., 2022), which could decrease the duration of future pinning points and speed up the retreat of Thwaites Glacier. However, while the forcings used in this study are idealised, they are based on the best available information of present-day extremes in observations (Dutrieux et al., 2014). These forcings also agree approximately with the time average of recent projections of future warming in the region (Naughten et al., 2023), which found linearly rising trends of ocean temperature in all tested climatic scenarios. The steady warm and cold forcings used in this study have an average temperature between 200-700 m depth of ~0.35 ℃ and ~-0.45 ℃ respectively. The warm case approximates the time-averaged temperatures at this depth over the similar trends projected in 100 year simulations of the Paris 1.5 ℃, Paris 2 ℃ and RCP 4.5 climatic conditions (Naughten et al., 2023), while the cold case approximates a cold historical state (Naughten et al., 2023). Another limitation of this study is the use of an ocean model that does not represent the evolution of oceanic conditions and sea ice on the wider Amundsen Sea continental shelf. As such, our model might lack important feedbacks such as increased ocean currents driven by ice-shelf meltwater bringing more CDW onto to shelf and driving higher melt rates (Kimura et al., 2017; Jourdain et al., 2017; Donat-Magnin et al., 2017). With an order of magnitude increase in ice shelf melting in our projections, such feedbacks would be substantial. In addition, the ocean simulation lacks the physical presence and effect of some freshwater sources, like sea ice and icebergs, which impacts the stratification of the water column, oceanic currents and the delivery of warm CDW to the base of ice shelves in the region (Bett et al., 2020). The lack of sea ice in particular could increase the heat content of CDW reaching the ice shelf bases, due to the lack of sea ice driven convective processes cooling the CDW layer (St-Laurent et al., 2015; Webber et al., 2017). However, it remains unknown how this cooling influence might vary over the coming centuries. These simulations additionally lack subglacial freshwater discharge, which at the grounding line has been found to increase ice shelf melting locally in previous ocean modelling studies (Nakayama et al., 2021), but overall, its effect is small in this region (Holland et al., 2023).

A key limitation of the model used in this study is the lack of calving-front retreat, which could impact ice dynamics, ocean conditions and total melting (e.g., Bradley et al., 2022; Joughin et al., 2021). As well as using a fixed temperature field, the model lacks an evolving damage field (Lhermitte et al., 2020). Therefore, the model may be sensitive to the time of initialisation, as this will determine the level of the damage that is applied for the entire forward simulation. However, we believe that these limitations cause our simulations to be a conservative estimate of Thwaites Glacier retreat for the cold and warm cases, as calving and evolving damage are only expected to enhance the retreat. Barring the existence of major calving- or damage-driven ice retreat over the next 180 years, our results suggest that the ocean-driven ungrounding of pinning points

will dictate the future SLR from the Amundsen Sea sector. However, strong sensitivity of ice-sheet projections to basal friction laws provides uncertainty to modelled ice retreat rates and mass loss, with the Weertman sliding law used in this study found to systematically predict the lowest ice mass losses (Brondex et al., 2019; Cornford et al., 2020). It should also be noted that 555 uncertainties in the accumulation field could potentially affect the modelled ice dynamics, including modelled SLR and grounding line retreat rates. In addition, the accumulation field used in this study is held steady and therefore any effects of future trends in this field are omitted, which could mitigate ocean driven ice loss to some extent (Edwards et al., 2021). Thus, this study focusses solely on dynamical ice loss driven by ocean melting.

560 All these ice dynamical limitations affect the no melt case results in this study. However, some of these limitations lead to specific additional caveats to the no melt case, where they may have the greatest impact. For example, the fixed ice front mask includes the current gaps between the east and west of Thwaites Ice Shelf, and these gaps cannot recover during the simulation. In the no melt case, where the ice shelf thickens and recovery of this damage should be possible, this could lead to an overestimation of SLR contributions from this hypothetical case.

565

## 5 Conclusions

This study presents, for the first time, 180-years of ice evolution in the Amundsen Sea sector of the West Antarctic Ice Sheet using a new synchronously coupled ice/ocean model, which includes full mass conservation in the coupled ice/ocean system, and an instantaneous response of melt rates to the evolving ice geometry. The coupled simulations were forced with idealised 570 warm and cold ocean conditions in the wider Amundsen Sea, and compared to each other, as well as to a zero ice-shelf melting case.

Even in the zero-melting case, the model predicts that the Thwaites and Smith areas lose ice mass during the simulations. This implies that the ice sheet model is initialised into an intrinsically unsteady state, so that in these simulations a tipping point 575 may have occurred in the past and we are now committed to further sea-level rise from this sector. However, when melting is activated in the coupled model, the rates of ice loss are much higher. This implies that ocean melting plays an important role in the future SLR contribution from this sector, though the difference between warm and cold scenarios is relatively modest, at only ~27% of the total SLR.

580 For Pine Island and Smith Glaciers, the rate of SLR remains relatively constant, at approximately 0.3 mm/yr and 0.1 mm/yr respectively, throughout the projections, leading to a linearly-increasing sea-level contribution. The Thwaites Glacier area provides a much larger sea-level contribution, and features an increasing SLR rate, which causes its sea-level contribution to increase approximately quadratically with time. The rate of SLR from Thwaites Glacier is closely controlled by the formation

and duration of isolated pinning points during the retreat of its grounding line. Ocean-driven melting is crucial in driving the ungrounding of these pinning points, by thinning the ice upstream, and this is the key mechanism by which future ocean conditions affect the SLR from this sector.

The coupled simulations show a large geometry-induced increase in total ocean-driven melting as Thwaites Glacier retreats and its ice shelf enlarges. This increased melting counteracts ice shelf thickening associated with thicker ice being advected across the deeper grounding line, which would otherwise cause the ice to ground downstream and arrest the retreat. Our simulations indicate large spatial and temporal variability in the melt rates at depth below 600 m. This variability will not appear within simple melting parameterisations.

Our results also suggest that accurate modelling of ocean-driven melting and ice response around pinning points, and accurate characterisation of the bed geometry and properties that lead to the formation of pinning points must be future research priorities. In addition, the further development and application of coupled ice/ocean models must be a priority, as it is difficult to envision how many of these results could have been achieved with parameterised ocean-driven melting, though further work is required to symmetrically compare the ice/ocean coupled model to the wide selection of melting parameterisations.

**Appendices**

**Appendix A: Ice model relaxation comparison**

The relaxation step that is applied to the ice model brings the rates of elevation change into much better agreement with observations, whilst resulting in only relatively small differences in the ice surface speed (Figure A1 e, f) and grounded ice thickness (Figure A1 b, c) compared to observations.

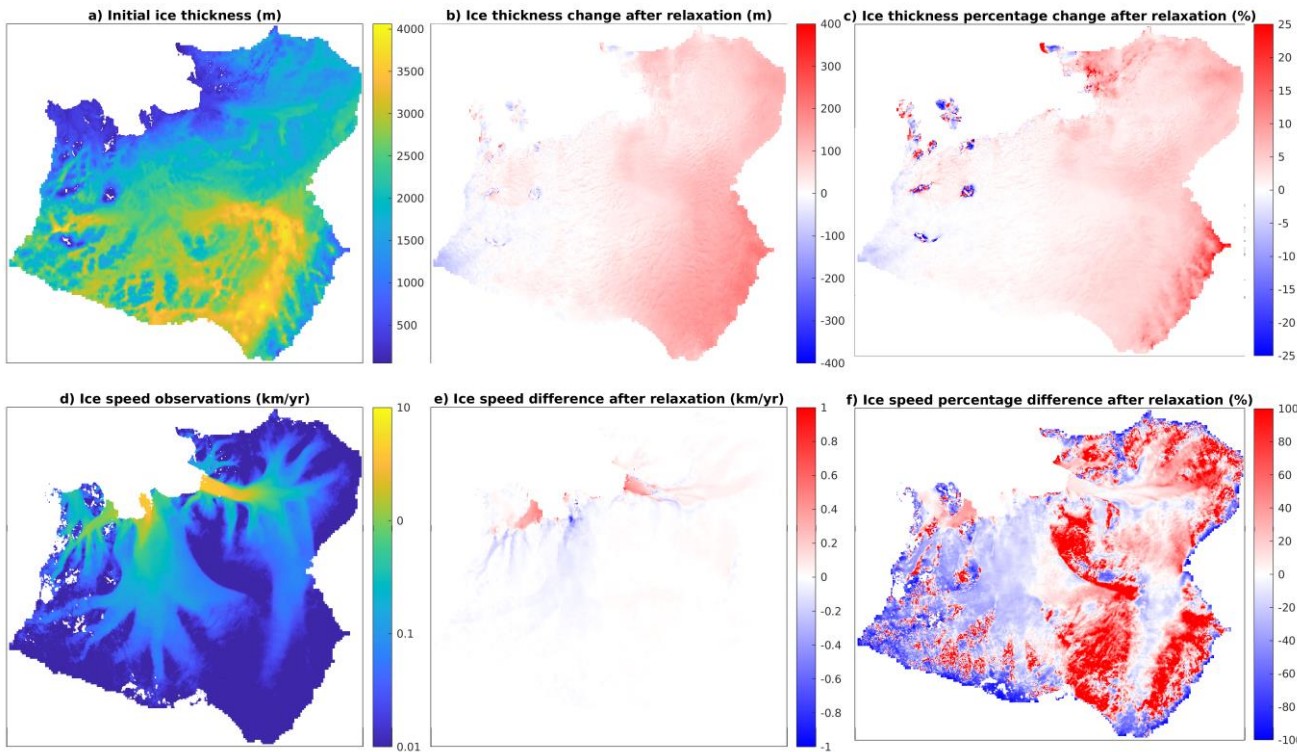


**Figure A1:** a) Initial ice thickness from the BedmachineV3 dataset (Morlighem et al., 2020; Morlighem, 2022) before relaxation. b) Changes in ice thickness from relaxation step. c) Percentage changes in ice thickness after relaxation step. d) Surface ice speed from MEaSUREs 2014/2015 (Mouginot et al., 2017a; Mouginot et al., 2017b). e) Model ice speed difference to observations after the relaxation step. f) Model ice speed percentage difference to observations after the relaxation step. All
plots shown for the full ice domain.

### Appendix B: Ice shelf melt rate tuning

In the initial setup of the ice model, we can calculate the implicit melt rate (IMR), which is the melt rate required to recreate observed present day surface elevation changes in the ice model, given initial ice model velocities and geometry (Arthern and Williams, 2017). In this model, we attempt to minimise initial coupling shock by reducing the mismatch between the ice model
IMR and the initial ice shelf melting field calculated from the MITgcm ocean model. To do so, we tune the dimensionless ice shelf melting drag coefficient in the three-equation formulation of melting parametrisation used in the MITgcm (Jenkins et al., 2010). We found that a choice of drag coefficient of 0.008 minimises the combined PIG and Thwaites ice shelves mismatch in total melt flux (Figure B1), while average ocean forcings are applied, with the thermocline placed at 700 m depth.

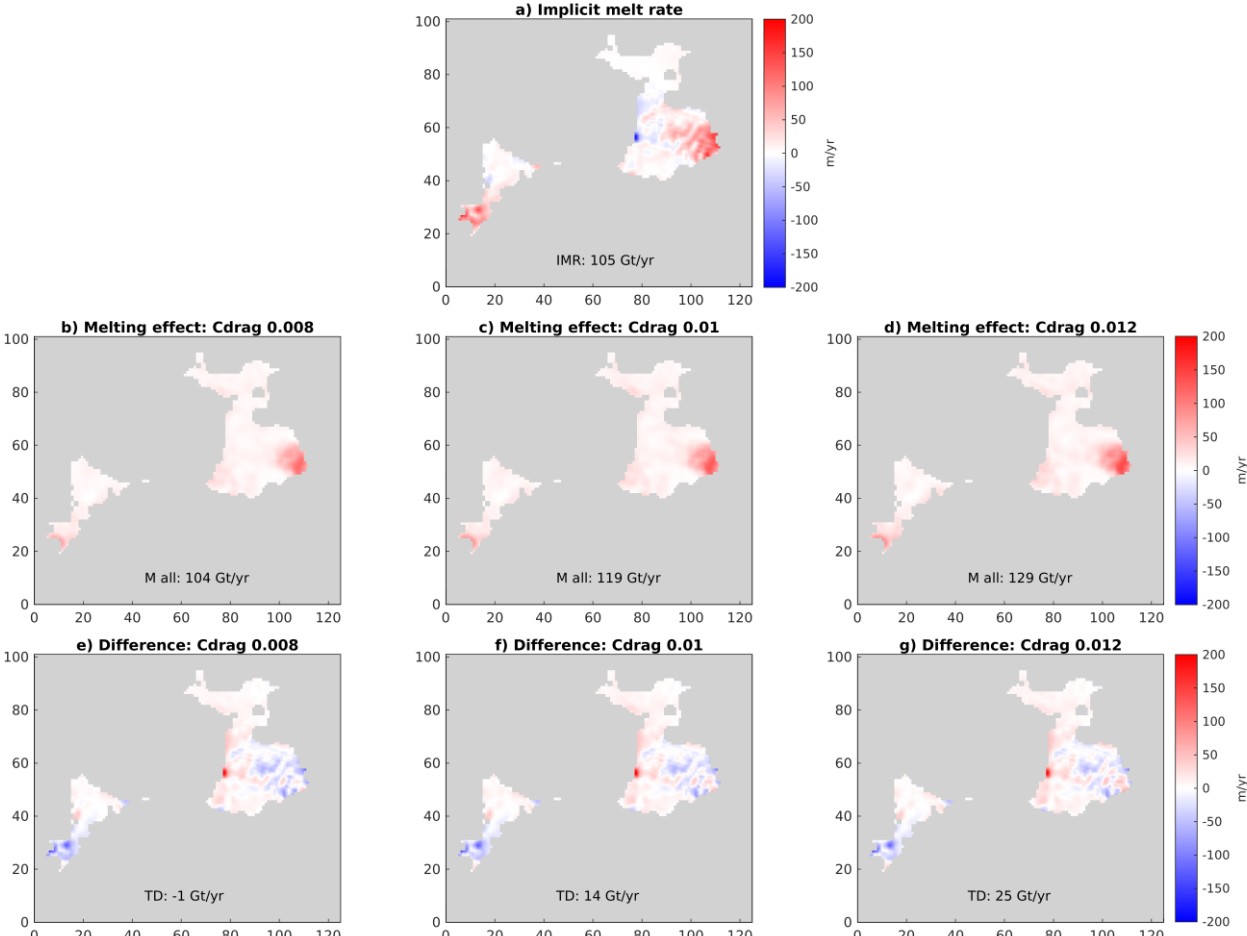

**FIGURE B1:** Area shown for PIG and Thwaites ice shelves. a) Initial implicit melt rate from the WAVI ice sheet model. b) Initial ice shelf melt rate from the MITgcm model for different values of the ice shelf melting drag coefficient as follows (and labelled): 0.008 (b), 0.01 (c), and 0.012 (d).. (e-g) Difference between initial implicit melt rate and initial ice shelf melting for the MITgcm model melt rates shown in panels (e)-(g) respectively.

## Appendix C: Initial bathymetry deepening

To ensure the bathymetry field from BedmachineV3 (Morlighem et al., 2020; Morlighem, 2022) has a specified minimum water column thickness under the initially floating ice area on the staggered Arakawa C-grid's velocity grid points, located on grid faces, we performed a `deepening' procedure, in which the bathymetry in areas with a water column thinner than a specific value is artificially deepened. This procedure was applied only to grid cells in which no ice basal sliding drag is applied in the initial state and occurs only once in a simulation before the WAVI ice sheet model is initialised and then relaxed. This is

because the bathymetry in ice shelf cavities is poorly known, while beneath the grounded ice sheet the bed is better known from radar soundings. We found this step is necessary because, without it, the initial PIG Ice Shelf cavity has an excessively thin water column near the grounding line. This leads to minimal ice shelf melting in this region (Figure C1c), which leads to a large mismatch with the calculated initial IMR (Figure B1). This, combined with the shallow bathymetry, leads PIG to

reground immediately at the start of the simulation, resulting in unrealistically low SLR rates from this region (Figure C1d).

Therefore, an initial deepening step was applied in order to prevent re-grounding and minimise any changes in SLR rate at the start of the simulation (Figure C1d). However, enforcing a uniform minimum water-column thickness everywhere would create a sudden step-change in the bathymetry at the initial grounding line location. Therefore, a taper is applied to the water column

thickness used in the deepening procedure, increasing the minimum thickness from a low value at the grounding line to its standard value over 6 km distance. After applying this initial deepening procedure, PIG does not reground at the start of the simulation. We note that the SLR rate from the Thwaites region (Figure C1a) is only minimally affected by the deepening procedure. While the deepening of the PIG water column is somewhat subjective, the deeper water column better reflects the sparsely available observations of the cavity near the grounding line (Dutrieux et al., 2014). Overall, the necessity of deepening

emphasises the importance of obtaining more detailed observations of the geometry of ice shelf cavities, beneath PIG in particular.

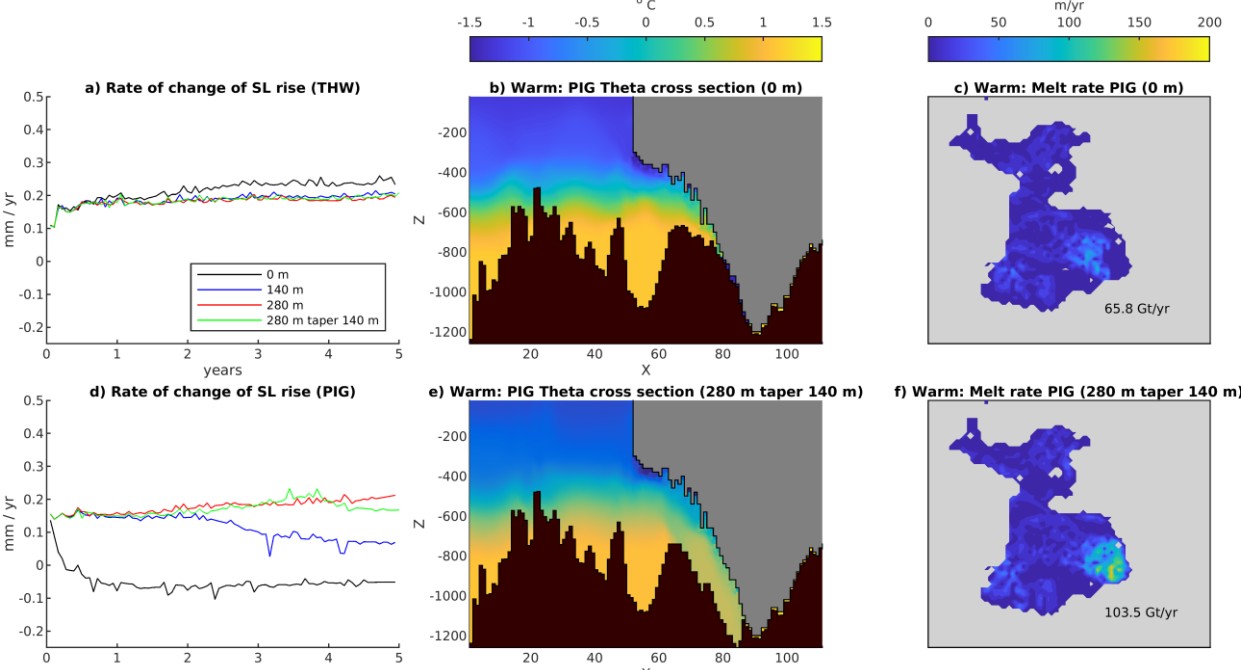

**FIGURE C1:** a) SLR rate of change for the Thwaites area for: zero deepening, 140 m deepening, 280 m deepening, 280 m deepening taper to 140 m over 6 km. b) Cross-section through PIG Ice Shelf (taken along the green line shown in Figure 2a) showing initial potential

temperature in the warm case in the zero-deepening setup. c) Initial melt rate over PIG Ice Shelf in the warm forcing case for the zero-deepening setup. d) Same as a) but for the PIG area. e) Same as (b), but for the 280 m taper 140 m deepening setup. f) Same as (c), but for the 280m taper 140 m deepening setup.

## Appendix D: SLR compared against integrated melt


A strong correlation is found when comparing the sea level contribution against the integrated melt for both forcing cases in the areas of Thwaites and PIG (Figure D1). However, no variation to the oceanic boundary forcing is applied during the simulations and hence only geometric induced changes in the ice shelf melt rates can occur. These strong correlations are due to the melt rate and rate of change of SLR being approximately constant from PIG's area and having an approximately linear 660 trend from Thwaites's area. The regression between the SLR contribution and the integrated melt for both the Thwaites and PIG areas are different for the two oceanographic forcings applied, which suggests that the ratio of mass loss between ice shelf melting and calving is different between the cases, with the warm case having a higher relative ice shelf melting.

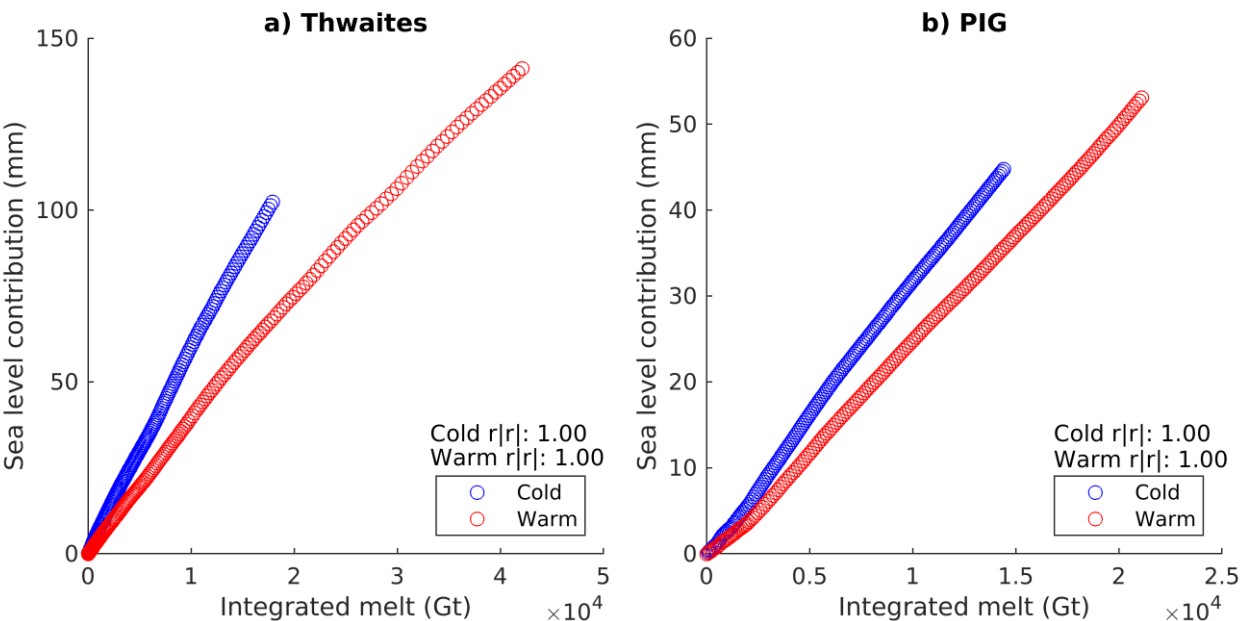

**FIGURE D1:** (a) Yearly points of sea level contribution plotted against integrated melt from the Thwaites area for the warm (red) and cold (blue) forced cases. (b) as in (a), but for the PIG area. In both, correlation coefficients are shown for the warm and cold cases.

## Code and data availability

The version of MITgcm used in this study is available at https://github.com/David-Bett4/MITgcm/tree/Coupling_wHoriz_lat_Pchild_new_divout. The version of WAVI used in this study is available at https://github.com/RJArthern/WAVI.jl/tree/MITgcm_coupling. Coupling scripts and input text files are available at https://github.com/David-Bett4/MITgcm_WAVI_coupling. The model output underlying the figures and calculations in this paper is available is available through the UK Polar Data Centre (DOI to be added) .

## Author contribution

DTB implemented coupling, set up ocean initial state, ran coupled simulations, performed analysis and led the manuscript. ATB and CRW set up initial WAVI ice model states and performed WAVI ice model relaxation simulations. RJA, PRH, CRW and DNG designed and supervised the project. All of the co-authors contributed to coupling design, experiment design and edited the manuscript.

## Competing interests

The authors declare that they have no conflict of interest.

## Acknowledgements

DTB, ATB, CRW, PRH, RJA were supported by the NERC Grant NE/S010475/1. CRW was partly funded by the MELT project, a component of the International Thwaites Glacier Collaboration (ITGC), with support from National Science Foundation (NSF: Grant no. 1739003 ) and Natural Environment Research Council (NERC: Grant no. NE/S006656/1). ITGC Contribution No. ITGC-108. This publication was supported by PROTECT. This project has received funding from the European Union's Horizon 2020 research and innovation programme under grant agreement no. 869304, PROTECT contribution number XX. This work used the ARCHER2 UK National Supercomputing Service (https://www.archer2.ac.uk). The authors thank the two anonymous reviewers for their careful reading of the manuscript and their helpful comments.

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
