# Peer review of "Coupled ice/ocean interactions during the future retreat of West Antarctic ice streams in the Amundsen Sea Sector"

_The Cryosphere, 2023_

## Author Comment (AC1)

**Response to Reviewers**

**Coupled ice/ocean interactions during the future retreat of West Antarctic ice streams**

David T. Bett, Alexander T. Bradley, C. Rosie Williams, Paul R. Holland, Robert J. Arthern and Daniel N. Goldberg

Black: Reviewer comments. Blue: Authors' response.  Where necessary to clarify our response, we have added proposed paper correction in italics, including new text as underlined where we are modifying existing text.

**Note to Editor and Reviewers:  Correction of a modelling error**

In addition to the changes in response to the reviewers, the simulations in this manuscript will be updated due to an error that was found in the model. We realised that the ice viscosity was not correctly averaged over the unevenly-spaced vertical sigma layers used by the WAVI ice sheet model. This led to an incorrectly low depth-average viscosity, due to over-weighting of the warm near-bed layers. Correcting this error required us to repeat the ice initialisation and relaxation procedures, as well as re-running all the projections. During this process, we also took the opportunity to implement a further change in that the bathymetry deepening described in the manuscript is now performed before initialisation and relaxation of the ice sheet model, rather than afterwards at the time of coupling to the ocean model.  This change is more of a modelling choice than a correction, but we believe the new approach is more defensible; the deepened bed is now taken into account when the inversion procedure matches the ice state to observations.

We have thoroughly investigated the impact of these two updates and found that they change the results in this manuscript quantitatively but not qualitatively. This will lead to minor revisions throughout the manuscript and figures. In particular, the resultant higher viscosity leads to projected ice changes that are slower, but qualitatively the same. For example, Figure R1 below shows SLR rates over Thwaites area for the original and updated simulations. Note the new longer time axis compared to the original manuscript. The jumps in the SLR rate, which occur upon pinning point ungroundings, are still present in the updated simulations, but occur at a later date. In the revised manuscript, the time over which the simulations evolve is extended to include a similar retreated area to the original simulations and to cover ungroundings from the same set of pinning points. The longer timescales will be reflected fully in the revised paper.

We would like to sincerely apologise to the reviewers and editor for the additional corrections that are caused by the ice model viscosity error, and the additional delay in the review process that resulted from having to rerun the simulations.

[Figure]

Figure R1 : Comparing the SLR rate from the original Thwaites area for the original simulations (a) and the updated ones after bug fixes (b), for the no melt (black), cold (blue) and warm (red) cases.

**Response to Reviewer 1**

Bett et al. presented the ice evolution of the Thwaites, Pine Island, and Smith Glaciers in the West Antarctic in a century scale using a new synchronously coupled ice/ocean model. Three couped simulations were conducted with warm and cold forcings and another one with no sub-shelf melting. They found the Thwaites Glacier provides a much higher sea-level contributions with a sea-level rise increasing approximately quadratically with time. The ice mass loss from Thwaites is closely dominated by the formation and duration of isolated pinning points and ocean-driven melting is the key driver behind the loss of pinning points. Overall, the manuscript is generally well written. However, the model setup section and some of the descriptions need more improvements.

We thank the reviewer for the time taken in reading and reviewing the manuscript and providing helpful comments and suggestions that will improve the manuscript.

Here are some general comments:

There are some details missing in the model setup section, especially about how the coupled model handles the grounding line movement. It is also not clear how the model facilitates the information exchange between the WAVI ice sheet models and the MITgcm STREAMCE every coupling period. Grounding line movement is a very important process in the coupled ice ocean models. However, the authors did not explain much about how they handle the grounding line movement in the coupled setup. You mentioned that (P5, L150) 'the WAVI drag field is passed to MITgcm to decide where melting can occur during each coupling period' and 'grounding-line retreat is accomplished naturally'. Do you mean the grounding line position is updated every ocean timestep and then passed to WAVI. Will the grounding line position be fixed in WAVI during each couple period? How do you decide the grounding-line should retreat in the ocean model?

We agree with the reviewer that the manuscript is currently unclear in its description on how the grounding line movement is handled and we welcome the opportunity to improve this. The manuscript will be revised to clarify that first we are explaining how the grounding line evolves in MITgcm only. The MITgcm package 'STREAMICE' is in control of when the grounding line is retreated in the ocean model due to previous coupling work between MITgcm STREAMICE and MITgcm ocean. This is fully documented in earlier papers (Jordan et al., 2018; Goldberg et al., 2018). Briefly, MITgcm solves the ocean free surface equation every ocean timestep in order to conserve mass. This naturally inflates the ocean beneath grounded ice wherever the ice thins so that its overburden pressure decreases sufficiently for the ocean to 'flood in' beneath the ice. A description of this will be added to the paper. In addition, we will expand the explanation of the coupled model procedure to explain that because to the WAVI timestep equals the coupling period, the WAVI state is fixed in between coupling periods (L176) and that the WAVI grounding line is only updated when MITgcm passes the new ice thickness calculated by STREAMICE (L180,182).

The boundaries between Thwaites, PIG, and Smith defined in Figure 1a are not reasonable to me. The following analysis on SLR contributions, melting and grounding ice area loss are all based on these boundaries. I would suggest using the basin boundaries to divide these three glaciers.

The areas used in this study (figure R2a) are modified from the basins in Zwally et al. (2012) (figure R2b), which will now be described in the manuscript. The areas used are motivated by the need to describe the qualitatively different future behaviour of Smith, Thwaites, and Pine Island glacier regions, and this requires us to modify the Zwally basins.

First, the Zwally basins do not separate out the glaciers surrounding Smith Glacier, so an approximate Smith area is defined by taking the far eastern edge of the Crosson Ice shelf down to Mount Takahe.

Second, the Zwally boundary between the PIG and Thwaites areas is inappropriate for future studies because the future Thwaites Glacier drains a significant area of ice that is allocated to PIG. Therefore the boundary between the two was edited by hand in order to address the otherwise mis-assignment of SLR contributions, melting and grounded ice area loss to PIG, as Thwaites retreats (Fig 3) past the original Zwally et al. (2012) boundaries in the simulations (Fig R2b). While this effect is relatively small it causes some qualitative issues with the results. For example, using the unmodified Zwally basins causes the PIG area to incorrectly display an increasing SLR rate (Fig R2e, f) and ice shelf melting (Fig R2 g, h).

This reasoning will be made clear in the manuscript when these areas are introduced, and the Figure 1 caption will mention that areas have been edited by hand.

*"These areas are edited from the Zwally et al. (2012) basins, where the Smith area has been separated out. In addition, the boundary between the PIG and Thwaites areas is edited by hand to address the otherwise mis-assignment of SLR contributions, melting and grounding ice loss to PIG, as Thwaites retreats (Fig 3)."*

[Figure]

Figure R2: a) Original areas used in this manuscript, b) Zwally et al. (2012) areas with Smith area still separated out. Using the original areas: c) THW rate of change of SLR, e) PIG rate of change of SLR, g) PIG ice shelf melting. Similarly using the Zwally et al. (2012) areas d), f), h). All figures are plotted using the updated simulations after bug fixes.

The authors suggested that the melting rates around the pinning points will be difficult to be captured without a coupled ice/ocean model. I would suggest running a separate ice-only model with parameterised basal melting and compare the difference.

We thank the reviewer for this suggestion and we agree that this would be an interesting comparison to make. However, there are many different types of parametrisations of basal melting and they generally lack a strong physical basis and perform poorly when considered in spatial and

temporal detail (Burgard et al., 2022). Furthermore, the parameterisations would have to be subjected to a comprehensive tuning effort in order to obtain a fair comparison. This means that robustly performing this comparison would require an extensive study of multiple parameterisations, and therefore we suggest that it would be out of the scope of the present manuscript, which already covers a substantial amount of material on the coupled model. This would however be very interesting work for a future study. We do however show that the ocean model creates a strong spatially variable melt field around these pinning points which would be hard to replicate with a basal melting parametrisation (Burgard et al., 2022). Text will be added to the discussion to explain this (L520).

*"Therefore, as parametrizations of basal melting generally lack a strong physical basis and perform poorly in spatial detail (Burgard et al., 2022), they may be expected to struggle to capture this spatial variability, affecting the evolution of these pinning points and the resultant SLR contribution rate from Thwaites. However, future work is required, running sets of ice only simulations, to perform an extensive comparison of the wide range of basal melt rate parametrizations."*

Specific Comments:

P3, L73: please provide the details (data name, version number, etc.) about the datasets (ice velocity and thickness change rates) for the inversion.

These details will be added to the manuscript.

P3, l79: It's not clear to me how you configure the englacial temperature in your ice model. I understand you cite Arthern et al., 2015 to cover the ice model setup but you should at least mention how the ice flow equation is represented in your ice model.

It will be made clearer in the manuscript which dataset is used for the englacial temperature (L85). In addition, a sentence will be added to state the ice flow law used in this study, which is the Glen flow law, along with the approximation to Stokes' equation that is solved (L77).

*"The ice rheology is described using Glen's flow law, with an exponent of n = 3. WAVI is a finite volume ice sheet model including a treatment of both membrane and simplified vertical shear stresses as described by Goldberg (2011)."*

P4, L97: eastern à western?

Thank you for spotting this and it will be fixed in the manuscript (L111).

P4, L105: When you say the initial melt rates, do you mean the melt rates you gor from the ocean only simulations at end of the 2 years spin-up?

That is correct, and will be made clear in the manuscript, by clarifying with "*after the 2 year ocean only spin-up*" (L120).

P4, L111: About the 'coupling shock', do you mean the sudden changes in the water fluxes across the ice/ocean boundary? Could you explain it in the text when you first mention it?

The 'coupling shock' is the response of the ice model to any mean offset between the MITgcm modelled ice shelf melt rates and those that are implicit in the WAVI initialisation. This mismatch is a model artefact. Given that projections are highly sensitive to their initial state, the response to any such shock could play a leading order role for many decades, contaminating the projections (Goldberg and Holland, 2022). This is why we have taken measures to minimise the coupling shock. This description will be expanded where it is first mentioned in the manuscript (L126).

*"For PIG and Thwaites ice shelves combined, this value produces the closest average match between the initial MITgcm ice shelf melt rates and those that are implicit in the WAVI initialisation (see Appendix A). This minimises the 'coupling shock' - the response of the ice model to any mismatch between these two fields -, which occurs when the coupled model simulation commences. Without this calibration, the ice sheet trajectory could be impacted, potentially for many decades (Goldberg and Holland, 2022), by the adjustment of the ice due to the transition from implicit initialised melt rates to arbitrarily different ocean model melt rates at the start of the simulation."*

P4, P114: what is the value derived from observation?

This sentence will be updated for the latest simulations and the value derived from observations will be included, which is 0.01 (L132).

P5, L 130: why do you set the subglacial layer to be 4 m thick. Is this a empirical value from previous studies?

The subglacial layer only exists in order to enable the expansion of ocean cells during ice retreat (Jordan et al., 2018; Goldberg et al., 2018). Therefore, this thickness is just required to be small compared to the ocean model vertical grid resolution, and 4 m is chosen for convenience. Text will be added to clarity this point (L 160).

*"In regions of ice which are not floating, a thin subglacial layer is specified in order to enable the expansion of the ocean column during grounding line retreat (Jordan et al., 2018; Goldberg et al., 2018). We set it to be 4 m thick but could have been set to any relatively small thickness compared to the ocean model vertical resolution. This small value has been previously demonstrated to have no impact on the evolution of the coupled system (Goldberg et al., 2018)."*

P5, L135: that in the 3D ocean grid that would never go afloat à that would never go afloat in the 3D ocean grid.

These will be swapped around as suggested.

P5, L138: how do you decide the couple domain is 'far enough'? Did you justify it based on a previous projection for this region?

The coupled domain was determined by running test simulations to find the extent of the modelled grounding-line retreat during the coming centuries. Text will be added to explain this in the manuscript (L172).

*"The coupled domain only needs to extend far enough inland to accommodate the grounding-line retreat occurring during a projection, which for this study was determined using test simulations."*

P5, L156: This information about bathymetry and initial ice geometry should be firstly mentioned in Sect. 2.1 and 2.2.

These will be moved as suggested, where the bathymetry deepening edits will be now introduced in Sect 2.2. The initial ice geometry is mentioned separately in Sect. 2.1.

P5, L160: what are 'velocity grid points'?

These will be clarified to be the Arakawa C-grid's velocity grid points (L138, 635).

P5, L161: when you talk about 'no ice basal drag', I understand you mean floating area. But in this study, you have two regions with basal drag: basal drag beneath the grounded ice and basal drag beneath the floating ice. Please clarify it across the text.

We thank the reviewer for spotting this source of potential confusion. The drag parameter used in the ice shelf melting will now be referred to throughout the manuscript as the "ice shelf melting drag" and the drag coefficient from the Weertman C sliding law will be referred to as the "ice sliding basal drag" or "sliding drag".

P7, L179: delete the first '(e)' here.

This will be deleted.

P8, L182: The colorbar for Fig3c melt rate is not good enough to show the increased basal melt near the new grounding line. Please adjust it to the visible range. Please also add the grounding lines in the caption.

The melt rate scale will be adjusted to better show the melt rates in Fig3c and the black/red grounding lines will be added to the caption description of this figure.

P9, L194: The Thwaites Glacier also shows a sign of deceleration by comparing the snapshots of 100 years and 125 years in Fig. 3b, which corresponds to the drop down in Fig 4b and 4c after year 100. I realise you mentioned this on P12L266-269, but I think you should at least point it out here and leave the explanation later.

This comment will be included as suggested for the new simulations (L242).

P9, L200: There are lots of noise on Figs. 4f and 4j but I did not see this noise in Fig. 4b. You ran the model for the whole domain and extract the rate of change rate of SL rise for each of the glaciers, right? Then why?

Some 'noise' is also present from the THW area, but it is harder to see due to this area being plotted on a much larger y-axis scale. Text will be added to highlight this (see below) (L 281).

*"In all three areas 'noise' is present in the SLR rates, though this is harder to see for the Thwaites area due to the larger y-axis scale."*

P10, L220: For Smith, it is closer to 0.15 mm/yr rather than 0.1.

This will be corrected and updated for the new simulations.

P10, L227: It's 'almost' 600 rather than 'over' 600. Actually, it is around 500 at end of 125 years.

This will be corrected and updated for the new simulations.

P10, L236-237: You mentioned that one of the limitations in this study was the idealised constant ocean forcings applied to the boundaries. Not just the decadal variability but also the climatology related changes under different emission scenarios in the future. How could you make the statement that 'the future SLR from this region is only weakly influenced by variations within the plausible range of ocean conditions'?

The oceanographic ranges in the CDW thickness used in this study represent the range in the observations. However, as the reviewer points out future oceanographic conditions in the regions could be outside of these ranges, hence this statement will be edited to be "within the observed range of present day ocean conditions" (L 294). However, while the forcings used in this study are idealised, they are based on the best available information of present-day extremes in observations and approximate the bounds of the latest projections of future warming in the region (Naughten et al., 2023). Text will be added to the discussion to highlight this point (L535).

*"However, while the forcings used in this study are idealised, they are based on the best available information of present-day extremes in observations (Dutrieux et al., 2014). These forcings also agree*

*approximately with the time average of recent projections of future warming in the region (Naughten et al., 2023), which found linearly rising trends of ocean temperature in all tested climatic scenarios. The steady warm and cold forcings used in this study have an average temperature between 200-700 m depth of ~0.35 °C and ~-0.45 °C respectively. The warm case approximates the time-averaged temperatures at this depth over the similar trends projected in 100 year simulations of the Paris 1.5 °C, Paris 2 °C and RCP 4.5 climatic conditions (Naughten et al., 2023), , while the cold case approximates a cold historical state (Naughten et al., 2023)."*

P17, L361: 'with only more modified CDW'? I think you mean 'only limited CDW'.

Pure CDW is only found outside the Amundsen shelf sea. The cooler water mass on shelf is modified by sea ice formation and glacial ice melting, and so cooler CDW is referred to as being more modified. This will be changed to "*with only more heavily-modified CDW*" (L437).

P19, L389: the SLR rate did not continue over the 125-year time period based on Fig. 4b if you are talking about the red line.

This will be updated and corrected for the new simulations.

P19, L407: 2 km mesh near the grounding line is not seen as a very high-resolution model. A coarser mesh near the grounding line may have underestimated the mass loss in the marine ice sheet systems.

A study that is currently under review (Williams et al., PREPRINT)(https://doi.org/10.21203/rs.3.rs-3405435/v1) uses a nearly identical configuration of the WAVI ice sheet model and investigates the effect of resolution on grounding line retreat and SLR contributions from the Amundsen Sea Sector, finding that lower resolutions actually lead to an overestimation of SLR contributions (Fig R3). That study finds that the sensitivity is minimal from 4km to 2km resolutions, suggesting the 2 km resolution used in this study is an appropriate resolution. This will now be described in the ice model methods of the manuscript (see below), and text will be added to discussion section to highlight that model resolution is an important consideration and may be one factor that plays a role in differences between studies in the literature (L 482).

*"We use a numerical model with a time step of 20 days and a 2 km horizontal resolution covering the whole Amundsen Sea sector domain shown in Figure 1a. This horizontal resolution is found to be appropriate in a recent study, currently under review, testing the impact of resolution on grounding line retreat and SLR contributions, with a nearly identical configuration of the Amundsen Sea sector (Williams et al., PREPRINT) (https://doi.org/10.21203/rs.3.rs-3405435/v1)."*

[Figure]

Fig R3: Taken from figure 2 in Williams et al. (PREPRINT)( https://doi.org/10.21203/rs.3.rs-3405435/v1). Sea level contribution (SLC) from the Amundsen Sea sector (mm) over 175 years for a range of model resolutions between 2km and 8km with prescribed average melt rates of 100 m yr$^{-1}$.

L20, P427: how about the subglacial freshwater discharge?

As suggested, a sentence will be added (shown below) to point out the potential impact of the lack of subglacial freshwater discharge, which could lead to underestimation of local ice shelf melt near the grounding line (L 551).

*"These simulations additionally lack subglacial freshwater discharge, which at the grounding line has been found to increase ice shelf melting locally in previous ocean modelling studies (Nakayama et al., 2021), but overall, its affect is small in this region (Holland et al., 2023)."*

L20, P438: The structure of discussion could be better organised. You're talking about another limitation here. It looks messy in the structure of the discussion. Why don't you talk about all these limitations together rather than separated by some other points like the paragraph above?

The discussion section will be rearranged as suggested with all limitations together and at the end of this section.

P21, L478: it would be interesting to see the differences by conducting the ice-only experiments with parameterised ocean-driven melting.

We agree that this would indeed be an interesting comparison to make. However, as mentioned above, due to the wide selection of possible melt rate parametrisations, an extensive comparison would be required, which we believe would be outside the scope of this study. A sentence has been added that this would be interesting future work and that more work is required in the comparison of coupled models and parameterised melting (shown with previous comment).

**References**

Burgard, C., Jourdain, N. C., Reese, R., Jenkins, A., and Mathiot, P.: An assessment of basal melt parameterisations for Antarctic ice shelves, The Cryosphere, 16, 4931-4975, 10.5194/tc-16-4931-2022, 2022.

Dutrieux, P., De Rydt, J., Jenkins, A., Holland, P. R., Ha, H. K., Lee, S. H., Steig, E. J., Ding, Q., Abrahamsen, E. P., and Schröder, M.: Strong Sensitivity of Pine Island Ice-Shelf Melting to Climatic Variability, Science, 343, 174-178, 10.1126/science.1244341, 2014.

Goldberg, D. N.: A variationally derived, depth-integrated approximation to a higher-order glaciological flow model, Journal of Glaciology, 57, 157-170, 10.3189/002214311795306763, 2011.

Goldberg, D. N. and Holland, P. R.: The Relative Impacts of Initialization and Climate Forcing in Coupled Ice Sheet-Ocean Modeling: Application to Pope, Smith, and Kohler Glaciers, Journal of Geophysical Research: Earth Surface, 127, e2021JF006570, https://doi.org/10.1029/2021JF006570, 2022.

Goldberg, D. N., Snow, K., Holland, P., Jordan, J. R., Campin, J. M., Heimbach, P., Arthern, R., and Jenkins, A.: Representing grounding line migration in synchronous coupling between a marine ice sheet model and a z-coordinate ocean model, Ocean Modelling, 125, 45-60, https://doi.org/10.1016/j.ocemod.2018.03.005, 2018.

Holland, P. R., Bevan, S. L., and Luckman, A. J.: Strong Ocean Melting Feedback During the Recent Retreat of Thwaites Glacier, Geophysical Research Letters, 50, e2023GL103088, https://doi.org/10.1029/2023GL103088, 2023.

Jordan, J. R., Holland, P. R., Goldberg, D., Snow, K., Arthern, R., Campin, J.-M., Heimbach, P., and Jenkins, A.: Ocean-Forced Ice-Shelf Thinning in a Synchronously Coupled Ice-Ocean Model, Journal of Geophysical Research: Oceans, 123, 864-882, https://doi.org/10.1002/2017JC013251, 2018.

Nakayama, Y., Cai, C., and Seroussi, H.: Impact of Subglacial Freshwater Discharge on Pine Island Ice Shelf, Geophysical Research Letters, 48, e2021GL093923, https://doi.org/10.1029/2021GL093923, 2021.

Naughten, K. A., Holland, P. R., and De Rydt, J.: Unavoidable future increase in West Antarctic ice-shelf melting over the twenty-first century, Nature Climate Change, 13, 1222-1228, 10.1038/s41558-023-01818-x, 2023.

Williams, C., Thodoroff, P., Arthern, R., Byrne, J., Hosking, J. S., Kaiser, M., Lawrence, N., and Kazlauskaite, I.: Calculating exposure to extreme sea level risk will require high resolution ice sheet models, Research Square [https://doi.org/10.21203/rs.3.rs-3405435/v1], PREPRINT.

Zwally, H. J., Giovinetto, M. B., Matthew, A. B., and Jack, L. S.: Antarctic and Greenland Drainage Systems, GSFC Cryospheric Sciences Laboratory, at http://icesat4.gsfc.nasa.gov/cryo_data/ant_grn_drainage_systems.php., 2012.

---

## Author Comment (AC2)

**Response to Reviewers**

**Coupled ice/ocean interactions during the future retreat of West Antarctic ice streams**

David T. Bett, Alexander T. Bradley, C. Rosie Williams, Paul R. Holland, Robert J. Arthern and Daniel N. Goldberg

Black: Reviewer comments. Blue: Authors' response.  Where necessary to clarify our response, we have added proposed paper correction in italics, including new text as underlined where we are modifying existing text.

**Note to Editor and Reviewers:  Correction of a modelling error**

In addition to the changes in response to the reviewers, the simulations in this manuscript will be updated due to an error that was found in the model. We realised that the ice viscosity was not correctly averaged over the unevenly-spaced vertical sigma layers used by the WAVI ice sheet model. This led to an incorrectly low depth-average viscosity, due to over-weighting of the warm near-bed layers. Correcting this error required us to repeat the ice initialisation and relaxation procedures, as well as re-running all the projections. During this process, we also took the opportunity to implement a further change in that the bathymetry deepening described in the manuscript is now performed before initialisation and relaxation of the ice sheet model, rather than afterwards at the time of coupling to the ocean model.  This change is more of a modelling choice than a correction, but we believe the new approach is more defensible; the deepened bed is now taken into account when the inversion procedure matches the ice state to observations.

We have thoroughly investigated the impact of these two updates and found that they change the results in this manuscript quantitatively but not qualitatively. This will lead to minor revisions throughout the manuscript and figures. In particular, the resultant higher viscosity leads to projected ice changes that are slower, but qualitatively the same. For example, Figure R1 below shows SLR rates over Thwaites area for the original and updated simulations. Note the new longer time axis compared to the original manuscript. The jumps in the SLR rate, which occur upon pinning point ungroundings, are still present in the updated simulations, but occur at a later date. In the revised manuscript, the time over which the simulations evolve is extended to include a similar retreated area to the original simulations and to cover ungroundings from the same set of pinning points. The longer timescales will be reflected fully in the revised paper.

We would like to sincerely apologise to the reviewers and editor for the additional corrections that are caused by the ice model viscosity error, and the additional delay in the review process that resulted from having to rerun the simulations.

[Figure]

Figure R1 : Comparing the SLR rate from the original Thwaites area for the original simulations (a) and the updated ones after bug fixes (b), for the no melt (black), cold (blue) and warm (red) cases.

**Response to Reviewer 2**

The manuscript "Coupled ice/ocean interactions during the future retreat of West Antarctic ice streams" by D. T. Bett and colleagues simulates the evolution of glaciers in the Amundsen Sea sector over a 125 year period using a synchronously coupled ice-ocean model. They find limited grounding line retreat and mass loss over Pine Island and Dotson-Crosson ice shelves but very large retreat on Thwaites glacier under conditions similar to warm ocean conditions observed over the past decade. The retreat rate varies spatially and with forcing conditions, and many pining points form during this retreat, suggesting a strong control of these pinning points and the importance of knowing the bathymetry precisely to accurately simulate the retreat of this glacier.

Overall, the manuscript is interesting and presents a detailed study of this region, however several aspects need to be improved and are sometimes misleading. In particular, the description of the models and processes used needs more clarification, as well as the limitations of this set-up. One particular aspect, since the pinning points play such an important role, is how the model handles parameters close to the grounding line: what is done if there are partially grounded cells, what is the friction and melt in this case, and what is the impact of the resolution in these areas on the retreat rate. These questions should be better investigated and discussed. One other potential problem is that there is no calving or ice front retreat and a thin layer of ice is "artificially" maintained; so melt continued under these thin parts, while there should not exist anymore. How does this impact the overall melt simulated and the simulations in general? Another missing part is the impact of the long spin-up of the ice model: how different is the configuration of the glaciers compared to observations after the 4000 year spin-up and how does this impact the possible retreat? Also, the forcing is highly idealized, which is clearly explained in the manuscript, but misleading in the title and abstract. There are a few places were previous studies are misrepresented. Also, it remains unclear what the coupled model brings to this study and what was learned that could not have been done with a standalone model. Finally, some figures need to be improved.

We thank the reviewer for their helpful comments and suggestions, which will certainly improve this manuscript. We quote below each of the specific points from this paragraph and answer them in turn.

1) "how the model handles parameters close to the grounding line: what is done if there are partially grounded cells, what is the friction and melt in this case, and what is the impact of the resolution in these areas on the retreat rate"

We agree with the reviewer that the decisions made on what happens near the grounding line are crucial, in regard to both ice shelf melting and basal sliding, so we will expand the description of how the model handles these aspects. Partially grounded cells are utilised in WAVI to better represent the grounding line, where the grounded ice fraction is used to proportionally apply the Weertman sliding drag coefficient, and this will be more fully explained (L83). The explanation of how the grounding line affects melting will also be expanded, including a description of what happens when a cell fully ungrounds (L196). As mentioned in our response to reviewer 1's comment, a study that is currently under review uses WAVI to investigate the effect of resolution on grounding line retreat and SLR contributions from the Amundsen Sea Sector in a nearly identical configuration to the present study (Williams et al., PREPRINT)(https://doi.org/10.21203/rs.3.rs-3405435/v1). This study finds that resolutions of greater than 4 km lead to an overestimation of grounding line retreat rates (Williams et al., PREPRINT)(https://doi.org/10.21203/rs.3.rs-3405435/v1) and this result will be

mentioned in the revised paper. Text will also be added to the discussion to include model resolution as a potential reason for differences between studies.

*"Partially grounded cells are utilised to better represent the grounding line, where the grounding fraction is used to proportionally apply the Weertman sliding drag coefficient."*

*"In addition, this means that if a cell becomes fully ungrounded in MITgcm, ice shelf melting only occurs once the coupling period finishes and a new one starts, which updates the ice thickness in WAVI, and subsequently passes back a new basal sliding drag field."*

2) "there is no calving or ice front retreat and a thin layer of ice is "artificially" maintained; so melt continued under these thin parts, while there should not exist anymore. How does this impact the overall melt simulated and the simulations in general?"

We agree that the fixed ice front in our simulations is a limitation of the model and we plan to address this in future studies. However, Figure 9 shows that most melting for Thwaites Glacier occurs at depth, below 600 m, and that the trend in the ice shelf melting discussed in the manuscript is occurring at depth. Additionally, only ~3% of Thwaites ice shelf total melting occurs on ice of thickness of 200 m or less at the end of the warm simulation, when Thwaites ice shelf is at its maximum extent. Therefore, we don't expect the fixed ice front, and the resulting large thin ice shelf, to have large impact on this result. Some text will be added to highlight this (L444). Additionally in combination with later comments, we will edit the range of the melt rate and ice thickness colour bars on Fig 3 in order to highlight that there are only low melt rates on thin ice for all of the ice shelves in model domain.

*"Most of the trend in ice shelf melting occurs in the deeper ice, with melting below 600 m peaking at an increase of ~30 times its initial value, which suggests that as the grounding line retreats, a strong increase in ice shelf base area below the thermocline controls total melting (Figure 9b). This confirms that the trend in melting does not result from the increasing ice shelf area associated with an artificially fixed ice front."*

3) "how different is the configuration of the glaciers compared to observations after the 4000 year spin-up and how does this impact the possible retreat?"

It is important to clarify the differences between the 4000 year relaxation and a normal forward simulation of the model. During the relaxation step the grounding line and the thickness of ice shelves are held fixed and the surface mass balance is set to equal the surface accumulation plus the observed thinning rate. This relaxation procedure removes artefacts in the ice thickness and brings the flux divergence into much better agreement with observations of accumulation and ice thickness change. This means that, at the end of the relaxation and start of the projection, the thinning is equal to the observed rate. This comes at the cost of the surface elevation and ice velocities agreeing slightly less well with observations. However, this relaxation step causes only relatively small differences in the ice surface speed (shown below in Figure R4 a, b) and grounded ice thickness (Figure R4 c, d) compared to observations, with the largest ice thickness changes occurring towards the edges of the full ice domain, which are dynamically less important regions (Figure R5 a-c). As a result, we expect this procedure to only have a small impact on the potential future retreat but lead to much better agreement of ice thinning rates at the start of the projection (Figure R4 c, d). The relaxation procedure is fully described in Arthern et al (2015). Text will be added to the paper explaining the relaxation's impact on the ice geometry and the purpose and consequences of this step (L84).

*"An ice model relaxation is then run for a set period of time (4000 years). During this relaxation the grounding line and the thickness of ice shelves remains fixed, but the grounded ice thickness is allowed to change (see Arthern et al. 2015 for full details). This brings the flux divergence into much better agreement with observations of accumulation and rates of ice thickness change but at the cost of the surface elevation and ice velocities agreeing slightly less well with observations."*

[Figure]

Figure R4: Histograms of the absolute difference between the initial model state and satellite observations on a logarithmic scale. For surface ice speed: (a) before relaxation, (b) after relaxation. For rates of elevation change: (c) before relaxation, (d) after relaxation. For grounded ice thickness: e) after relaxation, with the model matching observed ice thickness before relaxation by design. Data sets used: ice thickness from Bedmachine V3 (Morlighem et al., 2020; Morlighem, 2022), surface ice speed from Measure 2014/2015 (Mouginot et al., 2017a; Mouginot et al., 2017b), rate of elevation change from Smith et al. (2020).

[Figure]

Figure R5: a) Initial ice thickness from the BedmachineV3 dataset (Morlighem et al., 2020; Morlighem, 2022) before relaxation. b) Changes in ice thickness from relaxation step. c) Percentage changes in ice thickness after relaxation step. Both plots shown for the full ice domain.

4) "the forcing is highly idealized, which is clearly explained in the manuscript, but misleading in the title and abstract"

We agree that the ocean forcing is highly idealised, with the aim of clearly isolating the response of the system to wider climate forcing. However, these scenarios can be related to reasonable expectations for future changes in this region. In retrospect, we realise that this was very poorly explained in the submitted manuscript and we're very grateful to both reviewers for highlighting this. As detailed in the response to reviewer 1, our warm and cold scenarios, while derived from present day extreme observations, bracket the range of expected time-averaged ocean changes in this region, and this will be described in the revised paper. Our model simulations are projections of future retreat of the ice streams in the Amundsen Sea sector, and so we wish to retain that description in the title. Therefore we propose changing the title to "Coupled ice/ocean interactions during future retreat of West Antarctic ice streams in the Amundsen Sea sector".

5) "it remains unclear what the coupled model brings to this study and what was learned that could not have been done with a standalone model"

In this manuscript we highlight two main results, focussing on Thwaites Glacier: 1) The importance of ice shelf melting around pinning points in governing the rate of ice retreat and 2) a future increase in melting of the deep ice as the grounding line retreats, leading to a sea-level rise that is quadratic in time. Both of these features are critically dependent upon the details of the evolution of melting. For example, the ungrounding of pinning points is determined by the rate and pattern of ocean melting immediately upstream of the pinning point, and the melting of new deep ice is controlled by the inflow of CDW into the expanded cavities. Existing parametrisations of ocean melting are based on relatively crude representations of the relevant ocean physics and could not be trusted, a priori, to reliably reproduce these details of the melting. Thus, the use of a coupled model is essential to the conclusions of this study.

As detailed in the response to reviewer 1, the extent to which parameterised melting could reproduce some of these results is a very interesting question, but would comprise a substantial future research study in itself, due to the wide variety and known limitations of ocean melting parameterisations in current use (Burgard et al., 2022). We believe the present manuscript already represents a very substantial piece of work and sets the benchmark for future studies in that vein.

In retrospect this philosophy was not clearly presented in the submitted paper, and so we will modify the text and add text as appropriate to highlight what the coupled model brings to study, the limitations of parametrizations and why a coupled model is needed (see below with new text underlined) (L494, L520).

*"In this study we clearly show the importance of these mechanisms in a synchronous coupled model of the future Thwaites Ice Shelf, with strong spatial variability in ice shelf melting determining the duration of these pinning points and therefore their influence on SLR. Our use of a synchronously coupled model means variations in the ice thickness are instantly felt in the ocean model's melting calculation. This could impact the speed of the evolution of these pinning points, though further work is required to determine the impact of this coupling on the ice dynamics."*

*"Therefore, as parametrizations of basal melting generally lack a strong physical basis and perform poorly in spatial detail (Burgard et al., 2022), they may be expected to struggle to capture this spatial variability, affecting the evolution of these pinning points and the resultant SLR contribution rate from Thwaites. However, future work is required to perform an extensive comparison of the wide range of basal melt rate parametrizations."*

Specific points are listed below.

l.1: the title is misleading: given that the scenarios are highly idealized, it seems inappropriate to talk about "future retreat". Similarly, the work is done for the Amundsen Sea sector, which is much narrower than the West Antarctic ice streams. The title should be rephrased to better capture the study done.

As described above, our model simulations are projections of possible future retreat of the ice streams, and so we wish to retain that description in the title. We do accept the point about Amundsen Sea sector, however. Therefore we propose changing the title to "Coupled ice/ocean interactions during future retreat of West Antarctic ice streams in the Amundsen Sea sector".

l.19: I would have liked to see a sentence about calving or the ice front retreat in the abstract.

We will mention in the abstract that the simulations used in this study include a fixed ice front.

l.28: should be: "with Thwaites Ice Shelf ..."

We thank the reviewer for spotting this and it will be fixed (L28).

l.32-34: Thwaites ice shelf is rather unconfined, it might be worth mentioning it here.

This suggestion will be included in the sentence (L35).

l.48: I am surprised to see here "simple ocean melt parameterisations" being mentioned: studies such as Reese et al., 2020 use parameterizations that are as complex as can be with today's knowledge, so I am not sure what the authors suggest could be more complex than that.

We agree that this point was not clear, and we welcome the opportunity to clarify this. This was intended to highlight that ocean melting parametrisations are simple in comparison to a full ocean model, rather than that the parametrisation used in the study mentioned were simple compared to others. These parameterisations make many simplifications of the ocean physics known to be important. For example, the parameterisation in the study mentioned (PICO) contains no Coriolis force, ocean mixing parameterisation, barotropic ocean flow, or lateral variation in the direction parallel to the grounding line. The text will be expanded to clarify this (L49).

*"However, these studies use ocean melting parametrisations that contain simplifications of important ocean physics such as Coriolis force, ocean mixing parameterisations, barotropic ocean flow, and lateral variation in the direction parallel to the grounding line, and hence lack spatial variation in melt rates caused by differences in ocean velocity and temperature."*

l.51: "De Rydt and Gudmundsson, 2016" (and same in the rest of the manuscript)

This will be fixed throughout the manuscript.

l.55: "e.g., Goldberg …"

This will be fixed (L57).

l.69-80: What about the rheology of the ice?

A sentence will be added stating that the rheology of the ice is described using Glen's flow law with an exponent of n = 3 (L77).

l.75: What is the impact of the long spin-up? How similar/different are the geometry, velocity, etc. compared to the initial configuration before the spin-up? How does it impact of the simulations?

Please see response above.

l.79: What values are used for the accumulation? What is the impact on the simulation and especially the mass gain/loss since this is a first order control on sea level contribution.

The source of the data set used for the accumulation field is referenced in the manuscript, but this will be made clearer (L91). Text will also be added to the discussion section to highlight the possible impact of this choice and the use of a steady accumulation field, which impacts the modelled SLR and grounding line rates (shown below with new underlined) (L574). In general, we will revise the paper to clarify that the study focusses solely on the dynamical ice loss driven by ocean melting.

*"It should also be noted that uncertainties in the accumulation field could potentially affect the modelled ice dynamics, including modelled SLR and grounding line retreat rates. In addition, the accumulation field used in this study is held steady and therefore any effects of future trends in this field are omitted, which could mitigate ocean driven ice loss to some extent (Edwards et al., 2021). Thus, this study focusses solely on dynamical ice loss driven by ocean melting."*

l.69-80: This description is missing a description of what happens in the reason close to the grounding line. How is the grounding line included? How are partially grounded cells treated if there are some? How is the melt and the friction close to the grounding line? This information seems key given the role of the pinning points and should be better described.

Please see response above.

l.89: What is the impact of having no sea ice? How does it impact the ocean and in particular the stratification?

The impact of the lack of some freshwater sources, including sea ice, is discussed in the discussion section. Text will be added to expand this to include other potential impacts of the lack of sea ice, including that this could increase the heat content of CDW reaching the ice shelf bases, due to the neglect of sea ice driven convective processes cooling the CDW layer (L546). However, it remains unknown how this cooling influence might vary over the coming centuries.

*"In addition, the ocean simulation lacks the physical presence and effect of some freshwater sources, like sea ice and icebergs, which impacts the stratification of the water column, oceanic currents and*

*the delivery of warm CDW to the base of ice shelves in the region (Bett et al., 2020). The lack of sea ice in particular could increase the heat content of CDW reaching the ice shelf bases, due to the lack of sea ice driven convective processes cooling the CDW layer (St-Laurent et al., 2015; Webber et al., 2017). However, it remains unknown how this cooling influence might vary over the coming centuries."*

l.95: What period is used for these observations?

The years for the warmest and coldest observed years will be added, which are 2009 and 2012 for the warmest and coldest respectively (L110).

l.109: What is the form of the drag coefficient? Is it velocity dependent?

A sentence will be added to explain the effect of the ice shelf melting drag coefficient. This coefficient parameterises the ocean stress on the ice base as a function of the ocean model's upper layer velocities, in order to calculate turbulent ocean heat and salt fluxes for use in the melting calculation (shown below with new text underlined) (L123). Thus, the melting parameterisation is velocity dependent, but the drag coefficient itself is not.

*"We tune the dimensionless ice shelf melting drag coefficient in this parametrization to 0.008. This drag coefficient parameterises the ocean stress on the ice base as a function of the ocean model's mixed layer velocities, in order to calculate turbulent ocean heat and salt fluxes for use in the melting calculation (Jenkins et al., 2010)."*

l.119 and l.135 seem to be contradictory

The statement in l135 will be clarified to be the "ocean/STREAMICE grid" (L169).

l.127: Does it also need to exist anywhere there is ice since this is how the ice thickness is computed?

In this sentence 'MITgcm ocean grid' will be changed to the 'MITgcm grid' (L157), which does need to exist everywhere in the coupled domain, as in this domain the ice thickness is evolved by MITgcm. Edits will be added to explain this (L149) and to increase clarity (L146).

l.139: How do you ensure a smooth transition in the ice thickness and that it does not diverge between the two models over time?

The WAVI boundary cells outside of the coupled domain are passed to the MITgcm domain and held fixed while MITgcm runs, and then the thickness throughout the coupled domain is passed from MITgcm to WAVI after each coupling step (L184). The ice thickness in WAVI is fixed during each coupling period because its timestep equals the coupling period (L176). This procedure keeps the ice thicknesses in the two models and domains from diverging. Text will be added to include this explanation in the manuscript (see below with new text underlined).

*"The coupled model solution procedure is split into coupling periods, chosen to equal the WAVI timestep of 20 days, meaning that the WAVI model state is fixed in between coupling periods."*

*"The WAVI boundary cells outside of the coupled domain are passed to the MITgcm domain and held fixed while MITgcm runs, and then the thickness throughout the coupled domain is passed from MITgcm to WAVI after each coupling step. This procedure, in addition to WAVI being fixed during each coupling period, keeps the two models/domains ice thicknesses from diverging and ensures a smooth transition between the two domains."*

l.143: What is the impact of using shorter or longer time steps for the coupling?

The primary impact of having a shorter/longer coupling timestep is that it affects the fastest response time for which the ice velocities can respond to changes in buttressing in the coupled domain, where the ice thickness changes on the ocean timestep. A sentence will be added to make this point (see below) (L183).

*"Therefore, the choice of the length of the coupling period determines the fastest response time for which the ice velocities can respond to changes in buttressing in the coupled domain, where the ice thickness changes with the ocean timestep."*

l.153: It would be better to put this information (and detail it a lot more) in the description of the ice model.

These points about the interactions between melting and the grounding line involve the ocean model via the coupling framework and hence are best explained after these elements have been introduced. However, these points will be expanded to include what happens to ice shelf melting when a grid cell becomes fully ungrounded (L196). An extra sentence will be added to the ice model description outlining how the grounding line is represented and how the ice basal sliding drag is affected (L83). See response to similar comment from reviewer 1 for added text.

l.160: What does it mean "on velocity grid points"? Where are they?

This will be clarified to be the staggered 'Arakawa C-grid's velocity grid points', which are located on the grid faces (L138, L635).

l.162: What happens when the grounding line retreat and new floating cells are formed?

A sentence will be added to clarify that the bathymetry deepening is only performed once, before the WAVI ice sheet model is initialised (see below with new text underlined) (L139). In addition, edits will be added to Appendix B to increase the clarity of this step.

*"This procedure is only applied to cells with no ice basal sliding drag in the initial state and is only done once at the start of the simulation, before the WAVI ice sheet model is initialized and relaxed, rather than being an ongoing process."*

Fig.1: the green color for Smith is hard to see. Also it would be best to use different colors for the top left and the top right since these things are not related. It would be best to use a loop to describe the coupling in b).

The colours used in the figure will be updated as suggested to increase clarity and to remove any similarity in colours used between Fig 1a) and Fig1b). Figure 1b) was set up to show the order of the coupling and to show where time evolves in the simulations. This figure will include new labels denoting "Start coupling period = i", then a new second faded out step 1 box denoted with "Start coupling period = i + 1" in order to make clear that this is process is repeated.

Fig.2: it would be good to also put the mean melt rate on the figure. It is confusing to have positive numbers for the melt rate but negative numbers for the total melt. I am surprised but the shape and extent of Thwaites ice shelf, where does it come from? In the caption: should be "the ocean domain for …" and "melt ate under PIG"

We thank the reviewer for spotting this and the negative values used for the total melt will be changed to be positive in Fig.2. The shape and extent of Thwaites ice shelf comes from the Bedmachine V3 data set, where the minimum thickness of 50 m is applied to create the initial ice extent, which will be mentioned in the manuscript (L 89). However, the cross section shown is south-north rather than along flow lines, which will also now be mentioned in the figure caption.

*The caption will be fixed to "melt rate under PIG", but "ice/ocean domain" will be updated to "ice/ocean coupled domain" instead to follow the terms used in the manuscript.*

Fig.3: Unit for the second column is "m/yr" which is very surprising. The third column has columns very hard to see (everything is blue), maybe change the scale or use a log scale to make it easier to see. How does the front evolve? Are there regions where the ice becomes really thin and therefore have no ice? How does that impact the total melt?

*The "m/yr" axis label is an error and we thank the reviewer for spotting this. This will be changed to "km/yr". The colour range for the ice shelf melting plots in Fig. 3 will be adjusted as suggested to make the results clearer. In addition, the figure panels will be adjusted to maximise their size. The ice front is fixed in the simulations and there is a minimum ice thickness of 50 m. While the simulations do have artificially large and relatively thin ice shelves, cells that reach this minimum thickness in the simulations are only primarily located at the ends of Crosson and Dotson ice shelves, where only low melt rates of less than 10 m/yr occur. Therefore ice shelf melting on these cells specifically has minimal impact on total melting and is primarily present in an area which isn't the central focus of this study. More generally for melting on thin ice, the adjusted colour bar Fig 3 for ice shelf melting will be combined with an adjusted colour bar for ice thickness, which together will show that only low rates of ice shelf melting occur on the thinner ice. Text will be included to make this point (shown below) (L 245).*

*"The fixed ice front leads to a large and thin future Thwaites Ice Shelf, which may lead to artificially elevated melting overall, though only low ice shelf melt rates occur on the thinner ice (figure 3a,c)."*

l.191: How much retreat?

*A value will be added to represent the approximate North-South retreat over the simulation for the warm simulation shown in Fig. 3 (L 243).*

Fig.4: There is a large difference between the no melt case and the melt cases for Thwaites glacier compared to the other glaciers, this should be better discussed in the text (maybe around l.240). Caption: "(blue line), and warm (red line)". Maybe "Cumulative grounded area"

*Following the next comment we assume a typo and that the comment is 'There isn't a large difference between the no melt case and the melt cases for Thwaites Glacier'. In the PIG area, ice shelf melting prevents the ice shelf from thickening and re-grounding on the bathymetry bump below it. The buttressing provided by this re-grounding in the no-melt case leads to the large differences between the no melt and melting cases in this area. While there is smaller relative difference in the Thwaites area, melting does change it from a linearly increasing SLR contribution to a quadratic one. Text will be added to highlight and discuss these points for the latest simulations (L 298). Caption will be fixed, and subplot title edited.*

*"Additionally, there is a larger relative difference between the melting and no melting cases for the PIG area compared to Thwaites. In the PIG area, ice shelf melting prevents the ice shelf from thickening and re-grounding on the bathymetric ridge below it. In the no melt case, the ice shelf regrounds on this ridge; the buttressing provided by this re-grounding leads to the large differences between the no melt and melting cases for the PIG area (though this is dependent on the bathymetry deepening). While there is a smaller relative difference for the Thwaites area, the presence of melting still has a significant impact, changing the SLR contribution from linearly increasing to quadratic ."*

l.225-230: Some discussion about the role of the fixed ice front and the thin shelves that therefore keep melting would be important. I would also comment more on the difference between the melt and no melt scenarios for the different ice shelves (e.g., PIG very large while Thwaites is limited).

The trend in ice shelf melting is explored later in the manuscript in Section 3.3 and is found to be primarily caused by melting of ice below 600m, so the fixed ice front is the not the primary reason for the trend. As mentioned above, this will be further highlighted in that section (L 444) and that this is explored later will be highlighted when the trend in ice shelf melt rates is first mentioned (L 286).

As described above, we will expand the comparison of the melt and no melt cases. In addition, we will add also add a new paragraph at the end of the discussion, where we will expand upon our caveats to the no melt results in this study (L578).

*"All these ice dynamical limitations affect the no melt case results in this study. However, some of these limitations lead to specific additional caveats to the no melt case, where they may have the greatest impact. For example, the fixed ice front mask includes the current gaps between the east and west of Thwaites Ice Shelf, and these gaps cannot recover during the simulation. This may have the greatest impact in the no melt case, where the ice shelf thickens and recovery of this damage should be possible, and this could lead to an overestimation of SLR contributions from this hypothetical case."*

Fig.5: What is the bathymetry impacted by the scenario (panel d: bathymetry under warm retreat). Caption: "presence of isolated pinning points"

Fig 5d,5e shows the bathymetry/Weertman C values under the retreated area of the warm case. However, this figure will be edited to increase clarity by instead showing the bathymetry/Weertman C over the whole region under initially grounded ice, but with lines showing the final retreated extents for the warm and cold cases. Caption will be fixed.

l.275: Maybe add a sentence about the difference between flat and elevated regions, or between regions sloping inland vs downstream.

l.275 in the submitted manuscript refers to the end of a paragraph which compares the grounding line retreat rates between the warm and cold cases. These differences in retreat rates between the warm and cold simulations are due to the differing durations of the pinning points, and this will be now highlighted in the manuscript (L 386). However, text will be added in the previous paragraph to note that the bathymetry gets deeper inland, which could promote retreat more generally (see below).

*'The bathymetry in this area generally deepens inland, which could promote grounding line retreat (Weertman, 1974; Schoof, 2007).'*

l.279: "shallow" -> "shallower"

This will be fixed in the manuscript.

Fig.6: It's not very easy to see the differences in panels e and f since everything is blue. Caption: "The area shown in Figures b-f is shown with a black box in Figure 3."

The colour scale range for the pinning point duration points Fig 6(d-f) will be updated to use log scales to focus in on the time scales of the pinning points that are the focus of this study. The caption will be edited as suggested.

l.311: "pinning points – the time …"

This will be fixed as suggested in the manuscript (L382).

l.313: How much lower?

This will be quantified and will be included in the manuscript and updated for the new simulations.

l.314: Why does it increase the length? If everything retreats faster, it is not clear why it would cause this kind of changes.

This sentence will be edited, and the word 'length' will be removed.

"Reducing the duration of pinning points increases the intensity of periods of rapid ice acceleration and grounding line retreat."

Fig.7: Add missing titles on panels e, f, k, l, q, and r. The dark blue for panels n and o does not seem needed. It's a bit confusing to have the simulation years and the actual years. Maybe it would be better to start the caption with something like: "Evolution of conditions as grounding line retreats over pinning points." Before going into the details.

Titles will be added to all panels and the colour bar scaling for panels n and o will be adjusted to only show positive ice speed up. An introductory sentence will be added to the figure caption. The year labels in the subplot titles indicating time after the pinning point has formed, will be changed to '9 years later' and '18 years later' to aid clarity.

l.328: "show the ice geometry …"

This will be fixed in the manuscript (L 403).

l.340: "The resulting loss …"

This will be fixed in the manuscript (L 415).

Fig.8 caption: should be left and right instead of to/bottom. Add figures are every 25 years in the caption.

We thank the reviewer for spotting this and this will be fixed in Fig 8's caption. In addition, it will be included that the figures are taken every 25 years in the figure caption.

l.370: slope does not really seem to become shallower on Figure 8.

This statement will be clarified to be that the slope below 600 m gets shallower (L 447). This is best observed when comparing Fig 8b 75 years and 100 years in the original manuscript.

l.373-376: add numbers to this description

These descriptions will include values from the updated simulations as suggested (L450-452).

l.404: How does that differ from the effects of pinning points in the standalone model?

The sentence will be expanded to include that in the coupled model, ocean model ice shelf melt rates are used and it is these melt rates that determine the durations of pinning points and therefore their influence on SLR (L 494).

*"In this study we clearly show the importance of these mechanisms in a synchronous coupled model of the future Thwaites Ice Shelf, with strong spatial variability in ice shelf melting determining the duration of these pinning points and therefore their influence on SLR. Our use of a synchronously coupled model means variations in the ice thickness are instantly felt in the ocean model's melting*

*calculation. This could impact the speed of the evolution of these pinning points, though further work is required to determine the impact of this on ice dynamics."*

l.406: What is needed to correctly model these small features?

This statement will be expanded to highlight that these features need to be explicitly resolved in the ice model and that the spatial variability in ice shelf melting around them recreated in order to correctly model these small features (shown below with new text underlined) (L 501).

*" We have shown that high resolution coupled ice-ocean models are required to investigate the effect of pinning points on ice dynamics, as these small features need to be resolved in the ice model, and the strong spatial variability in ice shelf melting around them needs to be recreated. "*

l.412 ("very weak") and l.415 ("important buttressing") seem contradictory.

This will be clarified to state that the important buttressing in the future will come from future pinning points and not from the very weak buttressing provided by the lateral margins of the unconfined ice shelf (L 511).

l.434: "spatial variability": How does it compare with observations and parameterizations?

Text will be added to discuss that parameterizations are not expected to capture this spatial variability, but more work is required to test the wide selection of melt rate parametrizations available. In addition, a sentence will be added explaining the lack of observations around such features and their importance to improve modelling efforts (L520).

*" Therefore, as parametrizations of basal melting generally lack a strong physical basis and perform poorly in spatial detail (Burgard et al., 2022), they may be expected to struggle to capture this spatial variability, affecting the evolution of these pinning points and the resultant SLR contribution rate from Thwaites. However, future work is required to perform an extensive comparison of the wide range of basal melt rate parametrizations. Observations of ice shelf melt rates around such features with sufficient spatial coverage are currently lacking but are essential for improving future modelling efforts."*

l.439: How about ocean conditions?

This will be added to the possible effects of having the fixed ice front as suggested (L 564).

l.446-447: rephrase

This statement will be expanded and rephrased (L 574).

The discussion is missing about what really is different with the coupled model and with a synchronous coupling.

The question of how our results differ from a standalone ice model with parametrised melting is addressed in the responses above.

In addition, our use of a synchronously coupled model (evolving the ice thickness at the same time step as the ocean model) means variations in the ice thickness are instantly felt in the melting calculation. This could impact the speed of the evolution of these pinning points, though further work is required to determine the impact of this.

As outlined above, edits will be added to the manuscript to highlight these important points (L 495).

Also, it could be good to compare this study to the results of Urruty et al. (2022) or Reese et al. (2022) as they seem quite different and suggest relatively stable grounding lines around Antarctica.

The comparison of the results in this manuscript to other studies will be expanded, including comparing the simulation to the two studies suggested. The differences may arise through the different initialisation strategies adopted in our study and the two studies cited here. A paragraph will be included (shown below) (L 475), which discusses such uncertainty and reasons as to why studies may come to different conclusions, and that future work is required with the coupled model to reduce this uncertainty using larger ensembles will full parameter testing.

*"The SLR contributions in this study of 70 – 89 mm after 100 years for the Amundsen sector are within the uncertainties of previous ensemble studies (Edwards et al., 2021). However, the SLR and stability suggestions in this study do differ from recent studies, which found that the present-day geometry is not inherently unstable when starting from a stable starting position (Hill et al., 2023), and that the Amundsen Sea sector has not tipped yet (Reese et al., 2023). We suggest that these differences arise primarily through the different ice sheet model initialisation strategies adopted, which variously use a spin up period (Reese et al., 2023), data assimilation of ice velocities into a steady state (Hill et al., 2023), or assimilation of ice velocities and observations of unsteady thinning (present study). Also, the date of initialisation and differences in resolutions, datasets used and model physics may also play an important role. Larger coupled-model ensembles are needed to assess these aspects. Without this there is high uncertainty, for example, in SLR contributions and the timings of pinning point ungroundings. This study is designed to provide a small number of physically-advanced coupled simulations focusing on ice/ocean processes, rather than providing a larger but uncoupled set of predictions of future SLR contributions from the region."*

Additionally, the time at which the ice model is initialized will be expanded in the discussion, as something in which the model may be sensitive to due to the lack of evolving damage field.

*"As well as using a fixed ice temperature field, the model lacks an evolving damage field (Lhermitte et al., 2020. Therefore, the model may be sensitive to the time of initialisation, as this will determine the level of damage that is applied for the entire forward simulation."*

l.449: the forcing is really idealized so it is a bit pushing to call these projections, maybe simply evolution.

While the forcing is idealised, it is based on the best available information of present day extremes in ocean conditions (Dutrieux et al., 2014), and approximates the range of expected future ocean conditions from the latest oceanographic projections (Naughten et al., 2023). These are simulations of the future ice sheet state and so while we agree these are not 'predictions', we do think the word 'projections' is a reasonable descriptor. As described in previous comment responses, we will add text in the discussion highlighting this point. In addition, as previously described a new paragraph will be added to further discuss the uncertainty in the simulations and that this study only provides a small number of future simulations focusing on ice/ocean processes, rather than providing a comprehensive prediction of future SLR contributions from the region. Future work is required with ensembles of coupled model simulations with full parameter sweeps to reduce this uncertainty.

L.461: "For Pine Island …"

This will be fixed (L 598).

l.461: provide numbers

The approximate constant value SLR rate of PIG and Smith will be included for the latest simulations (L 598).

l.472: "temporal variability": it seems relatively constant on Fig.9c. Is it about the total or the area averaged?

This statement refers to the melt rate averaged over the ice area below 600 m, which will be clarified (L 609). Temporally the average melt rate below 600m ranges from 30 m/yr to 70 m/yr over the course of the simulation in the original manuscript, shown by the blue line in Fig 9c. This shows that during the simulation the average melt rate below 600 m can vary by 50%, which will be highlighted in the manuscript (L 452).

L.486: What is the form of the equation with drag coefficient? What is the unit?

The ice shelf melting drag coefficient is a dimensionless parameter, which will be stated when this parameter is first introduced in the manuscript and in Appendix A.

Fig.A1: It is confusing to see the difference compared to b) and then the difference compared to observations.

As suggested the figure will be updated so that the ocean model melt rates plots (Fig A1 c-d) are no longer shown as differences.

l.505: How do you deal with the very thin water column thickness that is created when the grounding line retreat? Why is this not a problem when it seemed a large problem in the initialization.

As clarified in a previous comment, the bathymetry deepening only occurs once, at the start of the simulation before the WAVI ice sheet model is initialised. The sentence on l.505 in the submitted manuscript refers to this deepening at the start of the simulation rather than an ongoing process that occurs during the simulation and only applies to the area in which is floating initially, where there is no subglacial layer present. As mentioned previously, sentences/edits will be added to Appendix B to improve the clarity of this step.

The seabed in ice shelf cavities is very poorly known, and this is the reason we deepen the shallow cavities initially. Beneath the grounded ice sheet, the bed is better known from radar sounding. Thus, we have no reason to deepen the bed beneath currently grounded ice (which becomes an ice shelf cavity as the ice retreats). Text will be added to make this point in Appendix B.

l.527: "strong correlation" but the slopes and relationships are different for the different oceanic conditions, so what does this suggest?

The regression between SLR contribution and the integrated melt in both the Thwaites and PIG areas are different for the two oceanographic forcings used. This suggests that the ratio between ice shelf melting and calving is different in the two oceanographic cases (the ice lost in SLR has to be either melted or calved). Text will be added to highlight this feature in Fig. C1 (L 676).

[revised manuscript text omitted]

---

## Referee Report (RR1)

Thanks for the detailed response to reviewers. The revised manuscript clearly addressed the questions and comments from reviewers. However, some of the descriptions need further improvement.

Here are some specific comments:

P3, L73: (Goldberg, 2011) → Goldberg (2011)

P3, L81: Measrues →  MEaSUREs

P3, L83: How do you decide the grounding fraction? Do you use sub-grid parameterisation method to decide the fraction? If yes, I think you need to further clarify it here.

P5, L132: a 0.01 value → a value of 0.01

L242: shows → show

L244: rapid melt rates → high melt rates

L254: are€?

L278: 'SLR than in the zero-melting case, increasing to ~0.3 mm/yr and ~0.1 mm/yr, in the warm case, respectively, ' → 'SLR in the warm case than in the zero-melting case, increasing to ~0.3 mm/yr and ~0.1 mm/yr, respectively '

L283: I'm curious what is causing the noise?

L293: 27% for all the three glaciers or you just talk about Thwaites here?

L371-372: the new added sentence is not quite clear to me. 'the last pinning point' → 'the last pinning point in group 'a''

L385-387: "with an average pinning point duration over warm/cold matching pinning locations of ~6 years" → "with an average pinning point duration of ~6 years over warm/cold matching pinning locations"

L387: ungrounding times → ungrounding timings

L484: "Without this" ?

L485: "the timings of pinning point ungroundings" → 'the timing when pinning point becomes ungrounded'

L493: recent observations of ??

L580-584: here you mention 'they may have the greatest impact' in the no melt case twice. Please rephrase it. Again, in this paragraph, I don't recall the gap between the east and west of Thwaites Ice Shelf. Have you mentioned it earlier in the manuscript? What do you mean "these gaps cannot recover during the simulation"? It is hard to follow what you discuss here without a context. Correct me if I have missed it somewhere.

---

## Author Response (AR2)

**Response to Reviewers**

**Coupled ice/ocean interactions during future retreat of West Antarctic ice streams in the Amundsen Sea sector**

David T. Bett, Alexander T. Bradley, C. Rosie Williams, Paul R. Holland, Robert J. Arthern and Daniel N. Goldberg

Black: Reviewer comments. Blue: Authors' response.  Where necessary to clarify our response, we have added proposed paper correction in italics, including new text as underlined where we are modifying existing text.

**Response to Reviewer 1**

Thanks for the detailed response to reviewers. The revised manuscript clearly addressed the questions and comments from reviewers. However, some of the descriptions need further improvement.

We thank the reviewer for the time taken in reading and reviewing the manuscript and providing helpful comments and suggestions that will improve the manuscript.

Here are some specific comments:

P3, L73: (Goldberg, 2011) -> Goldberg (2011)

This has been changed.

P3, L81: Measrues -> MEaSUREs

This has been changed.

P3, L83: How do you decide the grounding fraction? Do you use sub-grid parameterisation method to decide the fraction? If yes, I think you need to further clarify it here.

The sentence has been extended to including, 'using a sub-grid parametrization' along with the addition of references (L82):

*'Partially grounded cells are utilised using a sub-grid parametrization to better represent the grounding line (e.g., Arthern and Williams, 2017; Pattyn et al., 2006; Cornford et al., 2012; Seroussi et al., 2014), where the grounding fraction is used to proportionally apply the Weertman sliding drag coefficient.'*

P5, L132: a 0.01 value -> a value of 0.01

This has been changed.

L242: shows -> show

This sentence has been expanded for clarity:

*'However, as the simulation progresses, both sides of the ice shelf accelerate, reaching speeds of up to ~10 km/yr, but then the whole ice shelf shows signs of deceleration between 144 years and 180 years.'*

L244: rapid melt rates -> high melt rates

This has been changed.

L254: are€?

This has been fixed.

L278: 'SLR than in the zero-melting case, increasing to ~0.3 mm/yr and ~0.1 mm/yr, in the warm case, respectively, ' -> 'SLR in the warm case than in the zero-melting case, increasing to ~0.3 mm/yr and ~0.1 mm/yr, respectively '

This sentence has been updated with reviewer's 2 suggestion (L 263). The sentence's arrangement has been kept, as it makes this point about both melting cases and not just the warm case, but the warm case's values are given as an example.

*'In PIG and Smith areas, the rates of SLR with melting are approximately constant but are higher than the rates of SLR in the zero-melting case, increasing to ~0.3 mm/yr and ~0.1 mm/yr in the warm case, respectively'.*

L283: I'm curious what is causing the noise?

Text has been added to expand on this point.

*In all three areas 'noise' is present in the SLR rates, due to the instant effects of the evolving grounded ice fraction field, though this is harder to see for the Thwaites area due to the larger y-axis scale.*

L293: 27% for all the three glaciers or you just talk about Thwaites here?

This value is when considering SLR from all three regions, which has now been clarified in the text (L 280).

L371-372: the new added sentence is not quite clear to me. 'the last pinning point' -> 'the last pinning point in group 'a''

This has been changed.

L385-387: "with an average pinning point duration over warm/cold matching pinning locations of ~6 years" -> "with an average pinning point duration of ~6 years over warm/cold matching pinning locations"

This has been changed.

L387: ungrounding times -> ungrounding timings

This has been changed.

L484: "Without this" ?

This has been clarified to be 'Without these model ensembles.' (L473)

L485: "the timings of pinning point ungroundings" -> 'the timing when pinning point becomes ungrounded'

This has been changed (L474):

*'Without these model ensembles there is high uncertainty, for example, in SLR contributions and the timing of when pinning points become ungrounded.'*

L493: recent observations of ??

This has now been clarified to be 'recent observations of grounding-line retreat rates.' (L482).

L580-584: here you mention 'they may have the greatest impact' in the no melt case twice. Please rephrase it. Again, in this paragraph, I don't recall the gap between the east and west of Thwaites Ice Shelf. Have you mentioned it earlier in the manuscript? What do you mean "these gaps cannot recover during the simulation"? It is hard to follow what you discuss here without a context. Correct me if I have missed it somewhere.

Second mention of 'may have the greatest impact' has been removed (L 564). The gaps in the ice extent between the east and west of Thwaites ice shelf are now explicitly mentioned in the Methods section (L 88-89).

**Response to Reviewer 2**

The revision of the manuscript "Coupled ice/ocean interactions during the future retreat of West Antarctic ice streams in the Amundsen Sea Sector" is much improved compared to the initial submission, with extensive clarifications on the models and processes applied as well as additional discussion of the limitations of the study and how it compares to previous work.

We thank the reviewer for the time taken in reading and reviewing the manuscript and for providing helpful comments and suggestions that will improve the manuscript.

The main limitation of the new version is that the calculation of total melt includes regions that have thinned significantly and reached the threshold of 50 meters beyond which ice shelves won't thin further and for which the sub-ice shelf melt is actually not melting the ice shelves. While I understand this is a feature of the model that is common to many similar models, accounting for melt in these regions (where melt is actually not applied) is not correct, and these total melt values should be post-processed afterwards to correct it in regions where there is no more melt. The authors claim in their response that the impact is small, but the correct solution is to post-process that value afterwards.

The inclusion of melting on cells which have reached the minimum thickness has only a very small effect on the total melt timeseries for each area (Figure R1). Therefore, their inclusion or not has no material effect on the plots and conclusions taken from the total melting trends. In addition, it is important to clarify that melting is actually occurring in these cells. The melting is being calculated and the meltwater is cooling and freshening the ocean. The only limitation is that this melting is not permitted to thin the ice. This melting is therefore having an oceanographic effect, and it is also partly having a glaciological effect; while melting on these cells doesn't result in the ice thinning, it does stop the ice in these cells from thickening away from the minimum ice thickness value. These points are now discussed on lines 275.

*'In addition, while the effect of the inclusion is small, we do include ice shelf melting that occurs on ice that has reached the minimum thickness, as while this melting is not allowed to thin the ice*

[Figure]

*further it still has a partial glaciological effect, by stopping the ice from thickening. Furthermore, this melting still has an oceanographic effect in the simulation by cooling and freshening the ocean.'*

**Figure R1:** Timeseries of total melting shown for the cold (blue) and warm (red) cases, when including all grid cells (solid lines) and when excluding melt on grid cells at the minimum thickness (dashed lines). Timeseries are shown for the whole domain (a), the PIG area (b), the Thwaites area (c) and the Smith area (d).

Another thing that could be better discussed in the decrease in total melt and SLR found for Thwaites Glacier over the last ~30 years, to understand what causes these decreases at the end of the simulations in and warm case.

We thank the reviewer for pointing this out and we welcome the opportunity to expand the text to explain this feature of the simulation. Thwaites Glacier retreats on to a shallower and less slippery ridge towards the end of the warm case's simulation (as shown in Figure 5d), which corresponds with the reduction in SLR rate from the Thwaites area. Text has been added to explicitly point out this in the manuscript (L319-322), as shown below:

*'Towards the end of the warm case simulation, the grounding line experiences rapid retreat (Figure 5b) across a deep and slippery bed section (Figure 5d-e) before slowing down as it encounters a ridge of shallower and less slippery bed (Figure 5d-e), where it remains until the end of the simulation. These features explain the large variations in the Thwaites area SLR rate in the last ~50 years of the*

*simulation (Figure 4b). In particular, the retreat onto the shallower and less slippery ridge at the end of the warm simulation decreases the ice flux across the grounding line and corresponds with the Thwaites region's decreasing SLR rate during the last 25 years of the simulation.'*

In addition, as Thwaites retreats on to the shallower ridge, i.e. the bed depth at the grounding line increases, the ice shelf area reduces. Thus, a smaller area of ice shelf is exposed to the warmest water, reducing the ice shelf melting. Text has been added (L 424) to point out the decrease in melting at the end of simulation and text has been added to explain the reason why (L432).

*'Figure 9a shows the evolution of total melt flux from the Thwaites Ice Shelf in the warm case, for both the entire ice shelf and for ice below 600 m depth only, which is the thermocline depth in this case. For the majority of the simulation, the total melt flux from the entire ice shelf increases, but it decreases during the last 25 years.'*

*'However, towards the end of the simulation the groundling line retreats slowly onto a shallower ridge (Figure 5d), where the shallowing grounding line depth decreases the ice shelf area below 600 m, and subsequently decreases the total amount of melting below 600 m.'*

I continue to think that the title is misleading since the scenarios are highly idealized and do not represent "the future retreat" of that region.

We have discussed other options for the manuscript title, and we still believe that including the word 'future' is appropriate, as the ice model is initialized using present day ice velocities and rates of ice thickness changes. The oceanographic boundary conditions that are applied represent the present-day range in oceanographic conditions and are clearly explained in the abstract and manuscript. Therefore, while the simulations do not represent 'the future retreat' that is expected to occur, it does represent retreat that occurs in the future. The title has been edited to remove the word 'the':

*'Coupled ice/ocean interactions during  future retreat of West Antarctic ice streams in the Amundsen Sea sector'.*

Below are some technical comments line by line (lines refer to the manuscript including track changes):

l. 1: As mentioned above, I continue to think the title is misleading

Please see response above.

l.17: "that is approximately quadratic in time" -> "that evolves approximately quadratically over time"

This has been changed.

l.35: "has a unconfined" -> "has an unconfined": also if it is unconfined, it should be less impacted by changes in the ice shelf, so I am confused here

This has been changed. The sentence has been changed to 'largely unconfined' and we have added 'but' for clarity:

*'Of particular concern is Thwaites Glacier, which has a largely unconfined ice shelf, but whose current pinning point on its eastern ice shelf appears to be weakening and has been predicted to unground within decades (Wild et al., 2022).'*

l.48, l.56 and other: "e.g." -> "e.g.,"

This has been changed in all locations.

l.79: "basal sliding drag": I would use basal drag or basal sliding but not both (this comes up many times in the text)

This term was expanded out from 'basal drag' due to a reviewer suggestion to stop the potential confusion with the ice shelf melting drag coefficient used in the ice shelf melting calculations.

l.81: "Measures" -> "MEaSUREs"

This has been changed.

l.83: Add some references about the partial cell basal sliding

References have now been added to this sentence:

*'Partially grounded cells are utilised using a sub-grid parametrization to better represent the grounding line (e.g., Pattyn et al., 2006; Cornford et al., 2012; Seroussi et al., 2014), where the grounding fraction is used to proportionally apply the Weertman sliding drag coefficient.'*

l.85: "remains fixed" -> "remain fixed"

This has been changed.

l.85: You should add a figure showing the deviation of ice thickness and velocity from present in an appendix

A new appendix has been added (L 601) and referred to (L87) in regard to the changes of ice thickness and surface ice speed after the relaxation period of the ice model, when compared to present day observations.

l.110: "2009 and 2012" -> "in 2009 and 2012"

This has been changed.

l.110: "has temperature" -> "has a temperature"

This has been changed.

l.147: "to evolve" –> "to evolve continuously"

This has been changed.

l.161: "exists is" -> "is"

This has been changed.

l.168: "of Amundsen Sea" -> "of the Amundsen Sea"

This has been changed.

Fig.1 caption: "coloured regions" -> "coloured contours"

This has been changed.

l.231: Consider changing the title of section 3.1

The title for section 3.1 has been changed to 'Simulated evolution of the Amundsen Sea sector'.

Fig.3: Add a scale (like 100 km) on one of the subplots. Also, the log scale should be used for the velocity

These changes have been made to figure 3.

Fig.3 caption: "Starting" -> "Initial"

This has been changed.

l.240: I thought this pinning point was called the Eastern Ice Rise

To be consistent in terminology in the manuscript it is just referred to as a pinning point located on the current eastern Thwaites ice shelf.

l. 255: "above" -> "a-d" (twice)

This has been changed.

l.268: You should add that in the case of Pine Island there is no SLR contribution but instead some mass gain in the no melt case

New text has been added to make this point (L 254):

*'In the PIG area there is a negative SLR contribution for the zero-melting case (mass gain), though the contribution from PIG is dependent on the particular choice of bathymetry deepening that is implemented (Appendix B).'*

l.277: "obtain" is not clear

This sentence has been changed to increase clarity (L263).

l.293-296: How does that compare to previous studies?

This figure refers to the difference between warm and cold scenarios, as a fraction of the total SLR in the model for the whole domain, which is ~27% in this study. In the discussion section, we compare against a previous coupled model study (Seroussi et al., 2017) for the Thwaites area and find a larger sensitivity to a realistic range of ocean forcings, in the shorter and longer term. However, we feel that it would be misleading to compare the specific SLR percentages differences, as previous studies perform different experiments such as applying different ranges of oceanographic conditions, and varying increases in the temperature of the boundary conditions, rather than raising or lowering the thermocline as done in this study.

Fig.5: the purple lines are hard to see, maybe a different color or thickner lines would be better. I would also use thicker lines for grounding line positions

The contours have been changed to red in the subplots (a, b, d, e) in order to help with the clarity of these plots, while the contour for final extent of the grounding line retreat in the warm case has been changed to magenta in subplot (d , e).

l.329: "Weertman C basal sliding drag coefficient" -> "Weertman drag coefficient" or something a bit more natural

This has been changed to *'Weertman C drag coefficient'* as suggested (L 313).

l.329" "area of initially grounded ice" -> "ice area initially grounded"

This sentence has been updated:

*'Figures 5d and 5e show the bed depth and 'Weertman C' drag coefficient over the ice area that is initially grounded.'*

l.340-341: This is for specific areas and for the slowly retreating regions mostly. My first impression looking at the overall patterns in Fig.5 a and b is that they are very similar. Then looking in more details we start to see changes, but the overall retreat rate is similar suggesting this is not controlled by the ocean thermal forcing but by the geometry of the glacier.

This statement has now been clarified and expanded (L329):

*'There are very large percentage increases in grounding-line retreat rates in*  *two clear bands on the retreated area (Figure 5c)*, where two bands of slow retreat rates in the cold case are not present in the warm case.*'*

l.386: remove "warm/cold"

This has been removed.

l.398: remove "strong"

We believe the reviewer is referring to l. 445 in the tracked changes document. This word has been removed.

l.445-447: As mentioned above, I think it is not correct to include the melt applied in the regions not allowed to think because the thickness has reached the minimum imposed. I understand this is feature of the melt that cannot be changed, which is ok. But the calculation of the melt should be corrected or post-processed to only include it in the areas that are actually seeing the melt happening. The correction might be very small (which would be great so conclusions would not change), but at the moment it is adding some "random" melt never used to thin the ice, so I don't think this is correct, especially since it should be possible to calculate it afterwards without rerunning the entire simulation.

Please see response above.

Fig.9a and b: It would be good to talk more about the decrease in melt and area below 600 m in the last ~30 years of simulations, to explain why this is happening after a long period of regular increase.

Please see response above.

l.469: "over 160 years of the time period" -> "over the first 160 years"?

This has been changed.

l.473: "sooner": sooner than what?

This is referring to the simulations in the previous ice only study and this text has been added to clarify this: 'than in the simulations of this previous study.' (L 449).

l.493: mention what observations you are talking about

This has now been clarified to be 'recent observations of grounding-line retreat rates.'(L482)

l.517: "increasing ice area at depth" is not clear (the area at depth is not really increasing since it was always there even if grounded). The following explanation on l.518 is much more clear so maybe remove this first part.

First part has been removed as suggested (L495).

l.554f: "affect" -> "effect"

This has been changed.

l.564: remove "ice shelf"

This has been removed.

l.567: "determine level" -> "determine the level"

This has been changed.

l.572: "sensitively" -> "sensitivity"

This has been changed.

l.581: remove comma after mask

This has been removed.

l.583: remove "in the region"

We believe that this comment refers to l.593. This has been removed.

References

Arthern, R. J. and Williams, C. R.: The sensitivity of West Antarctica to the submarine melting feedback, Geophysical Research Letters, 44, 2352-2359, https://doi.org/10.1002/2017GL072514, 2017.
Cornford, S. L., Gladstone, R. M., and Payne, A. J.: Resolution requirements for grounding-line modelling: sensitivity to basal drag and ice-shelf buttressing, Annals of Glaciology, 53, 97-105, 10.3189/2012AoG60A148, 2012.
Pattyn, F., Huyghe, A., De Brabander, S., and De Smedt, B.: Role of transition zones in marine ice sheet dynamics, Journal of Geophysical Research: Earth Surface, 111, https://doi.org/10.1029/2005JF000394, 2006.
Seroussi, H., Morlighem, M., Larour, E., Rignot, E., and Khazendar, A.: Hydrostatic grounding line parameterization in ice sheet models, The Cryosphere, 8, 2075-2087, 10.5194/tc-8-2075-2014, 2014.
Seroussi, H., Nakayama, Y., Larour, E., Menemenlis, D., Morlighem, M., Rignot, E., and Khazendar, A.: Continued retreat of Thwaites Glacier, West Antarctica, controlled by bed topography and ocean circulation, Geophysical Research Letters, 44, 6191-6199, https://doi.org/10.1002/2017GL072910, 2017.